# The human cytoplasmic dynein interactome reveals novel activators of motility

William B Redwine[1,2†], Morgan E DeSantis[1†], Ian Hollyer[1‡], Zaw Min Htet[1,3], Phuoc Tien Tran[1], Selene K Swanson[4], Laurence Florens[4], Michael P Washburn[4,5], Samara L Reck-Peterson[1,6*]

[1]Department of Cellular and Molecular Medicine, University of California, San Diego, United States; [2]Department of Cell Biology, Harvard Medical School, Boston, United States; [3]Biophysics Graduate Program, Harvard Medical School, Boston, United States; [4]Stowers Institute for Medical Research, Kansas, United States; [5]Department of Pathology and Laboratory Medicine, The University of Kansas Medical Center, Kansas, United States; [6]Division of Biological Sciences, Cell and Developmental Biology Section, University of California, San Diego, United States

*For correspondence: sreckpeterson@ucsd.edu

†These authors contributed equally to this work

Present address: ‡Feinberg School of Medicine, Northwestern University, Chicago, United States

**Abstract** In human cells, cytoplasmic dynein-1 is essential for long-distance transport of many cargos, including organelles, RNAs, proteins, and viruses, towards microtubule minus ends. To understand how a single motor achieves cargo specificity, we identified the human dynein interactome by attaching a promiscuous biotin ligase ('BioID') to seven components of the dynein machinery, including a subunit of the essential cofactor dynactin. This method reported spatial information about the large cytosolic dynein/dynactin complex in living cells. To achieve maximal motile activity and to bind its cargos, human dynein/dynactin requires 'activators', of which only five have been described. We developed methods to identify new activators in our BioID data, and discovered that ninein and ninein-like are a new family of dynein activators. Analysis of the protein interactomes for six activators, including ninein and ninein-like, suggests that each dynein activator has multiple cargos.

## Introduction

Microtubules and their motors are the primary means of long-distance intracellular transport in humans and many other eukaryotic organisms. Mutations in the transport machinery cause both neurodevelopmental and neurodegenerative diseases (*Lipka et al., 2013*). Microtubules are polar structures, with dynamic 'plus' ends typically found near the cell periphery and 'minus' ends anchored in internal microtubule organizing centers. Dynein motors move towards the microtubule minus end, whereas most kinesins move in the opposite direction. The human genome contains 15 motor domain-encoding dynein genes (*Vale, 2003*), but only cytoplasmic dynein-1 (DYNC1H1; 'dynein' hereafter) is involved in long-distance, minus-end-directed transport in the cytoplasm. Dynein transports dozens of distinct cargos including organelles, ribonucleoprotein complexes, proteins and viruses (*Kardon and Vale, 2009*). A major outstanding question in the field is to understand how dynein achieves temporal and spatial specificity for cargo interactions.

Most cytoskeletal motors that transport cargos over long distances in cells are processive motors, capable of taking multiple steps along their track. While dimers of the *S. cerevisiae* dynein heavy chain move processively in the absence of cofactors (*Reck-Peterson et al., 2006*), mammalian

dynein requires the 1.1 MDa dynactin complex and a coiled coil-containing activator ('activator' hereafter) for robust processive motility (*McKenney et al., 2014*; *Schlager et al., 2014*; *Trokter et al., 2012*). Activators have a second function; they also link dynein/dynactin to cargo (*Figure 1A*) (*Cianfrocco et al., 2015*).

Currently, there are five proteins that likely function as dynein activators. The activators BICD2 and HOOK3 have been definitively shown, using purified components, to activate dynein/dynactin motility in vitro (*McKenney et al., 2014*; *Schlager et al., 2014*; *Schroeder and Vale, 2016*). HOOK1, Spindly (SPDL1), and RAB11FIP3 are also likely activators based on their ability to co-purify and co-migrate with dynein/dynactin in in vitro motility assays (*McKenney et al., 2014*; *Olenick et al., 2016*). Other proteins may be activators based on their homology to BICD and HOOK family activators, including BICD1, BICDL1, BICDL2, and HOOK2 (*Hoogenraad and Akhmanova, 2016*; *Simpson et al., 2005*). Known and predicted activators all contain long stretches of predicted coiled coil, but share very little sequence homology across activator families (*Gama et al., 2017*); currently it is not possible to identify activators based on sequence alone. Central to understanding how dynein performs so many tasks is to determine if it has additional activators.

Here we used new proteomics tools to address major unanswered questions about dynein-based transport. What is the dynein protein interactome? How many activators does dynein have in a given cell type? Which cargos do activators link to? Does each cargo have its own activator? To answer these questions, we used proximity-dependent labeling in living human cells. Traditionally, protein-protein interaction discovery using immunoprecipitation followed by mass spectrometry has been confined to relatively stable interactions. However, recently developed methods such as BioID (*Roux et al., 2012*) and APEX (*Rhee et al., 2013*) allow the discovery of weak and short-lived interactions in living cells, in addition to more stable interactions. The BioID method relies on expressing a protein of interest fused to a promiscuous biotin ligase that releases activated biotin-AMP in the absence of substrate (*Roux et al., 2012*). Biotin-AMP covalently modifies the primary amines of proximal proteins within a nanometer-scale labeling radius (*Kim et al., 2014*). Biotinlyated proximal proteins are identified by isolation with streptavidin followed by tandem mass spectrometry (MS/MS). For example, this approach has been used to map protein interactions at human centrosomes and cilia (*Gupta et al., 2015*), focal adhesions (*Dong et al., 2016*) and the nuclear pore (*Kim et al., 2014*).

Using these methods we describe the human dynein/dynactin interactome. We also developed methods to identify dynein activators within these datasets and identified two new activators that constitute a novel activator family. Finally, to determine the candidate cargos of six distinct activators we elucidated their individual interactomes. Our results suggest that each dynein activator has multiple cargos. We propose that activators provide the first layer of defining cargo specificity for cytoplasmic dynein, but that refinement of cargo selection will require additional factors.

## Results

### Identification of the dynein/dynactin interactome

To identify the human dynein/dynactin interactome, we began by biochemically characterizing dynein and dynactin subunits fused to BioID that were stably expressed in HEK-293 cells. The 1.4 MDa dynein holoenzyme is composed of dimers of heavy chains (HC; DYNC1H1), intermediate chains (IC1 or IC2; DYNC1I1 and 2), light intermediate chains (LIC1 or LIC2; DYNC1LI1 and 2), and three types of light chains: Roadblock (RB; DYNLRB1 and 2), LC8 (DYNLL1 and 2), and TCTEX (DYNLT1 and 3) (*Figure 1A* and *Figure 1—figure supplement 1*). We first generated a cell line stably expressing IC2 with C-terminal BioID G2 ('BioID' here) (*Kim et al., 2016b*) and 3×FLAG tags. Immunoprecipitations confirmed that IC2-BioID was incorporated into the dynein/dynactin complex (*Figure 1B—E*). Gel filtration analysis of IC2 immunoprecipitates revealed that 51% of the BioID-tagged IC2 was incorporated into the dynein complex (*Figure 1E*). We obtained similar results when BioID was fused to the C-terminus of the p62 dynactin subunit. The stably expressed p62-BioID-3×FLAG subunit incorporated into the dynactin complex as shown by immunoprecipitations (*Figure 1F*), and gel filtration analysis of these immunoprecipitatates revealed that 47% was incorporated into the high molecular weight dynactin complex (*Figure 1G*).

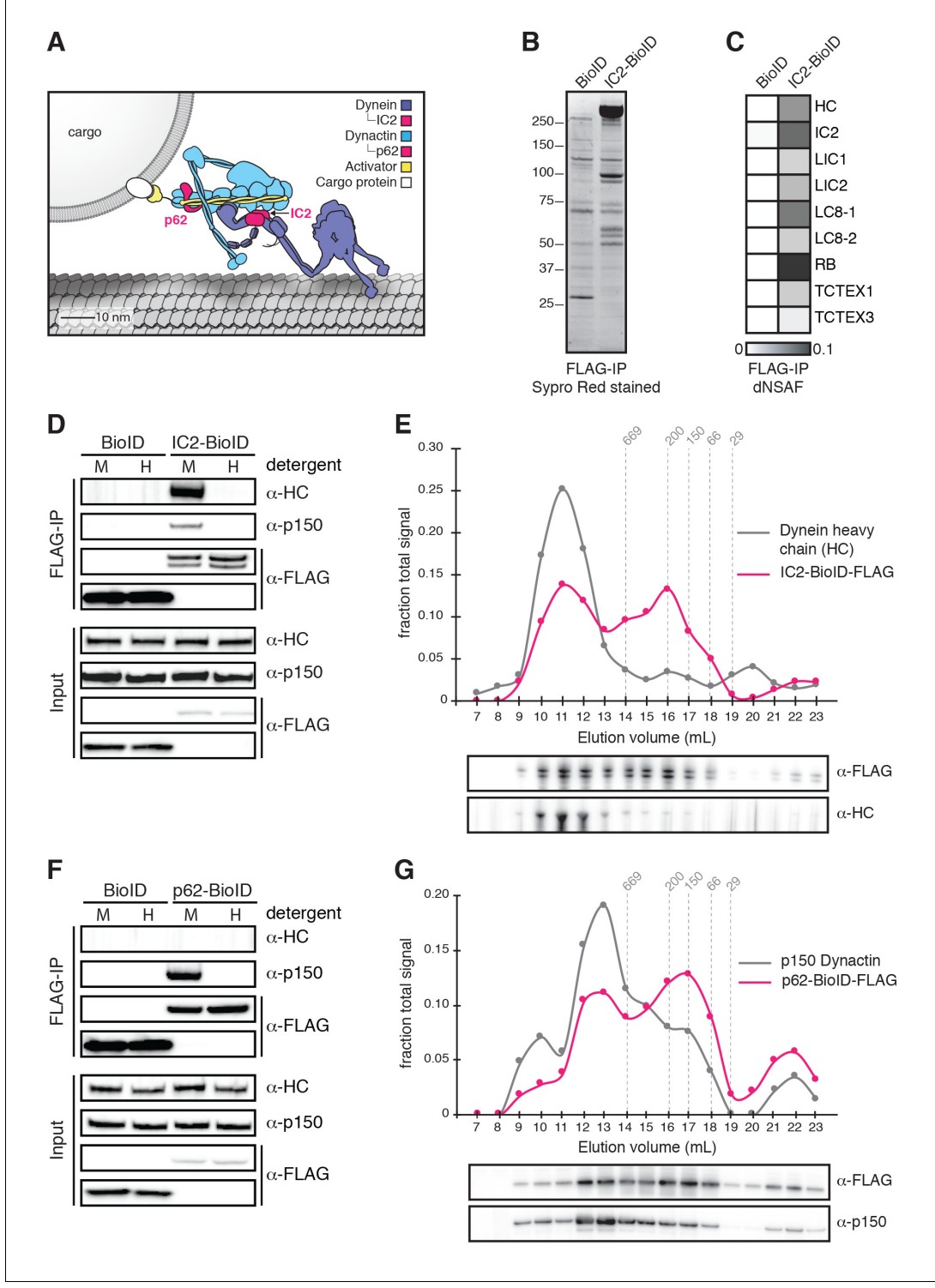

**Figure 1.** Validation of BioID-tagged dynein and dynactin subunits. (**A**) A cartoon of the dynein/dynactin/activator complex based on cryo-EM structural studies (*Chowdhury et al., 2015*; *Urnavicius et al., 2015*) with proteins drawn to scale. (**B**) BioID-3×FLAG or IC2-BioID-3×FLAG were immunoprecipitated from stable HEK-293 cell lines using α-FLAG antibodies and eluted using FLAG peptide. A Sypro Red stained SDS-PAGE gel of the immunoprecipitates is shown. (**C**) MS/MS analysis of the immunoprecipitates from (**B**). Core dynein subunit dNSAF (distributed normalized spectral abundance factor) (*Zhang et al., 2015*) values are displayed as a gray scale heat map. (**D**) Immunoprecipitations were performed as in (**B**) with mild (M) or harsh (H) detergent conditions (see

*Figure 1 continued on next page*

*Figure 1 continued*

Materials and methods). Harsh detergent conditions disrupt IC2 incorporation into the dynein/dynactin complex as shown by Western blots with α-HC and α-p150 (dynactin subunit) antibodies. (E) IC2-BioID-3×FLAG was immunoprecipitated from a stable HEK-293 cell line using α-FLAG antibodies and fractionated by gel filtration FPLC chromatography. Fractions were analyzed by Western blotting with α-FLAG and α-HC antibodies. The signal intensity for IC2-BioID-3×FLAG (magenta) and HC (gray) in each fraction is plotted as a fraction of the summed intensity of all fractions. The elution volumes of molecular weight standards are indicated (dashed lines). (F) BioID-3×FLAG or p62-BioID-3×FLAG were immunoprecipitated from stable HEK-293 cell lines using α-FLAG antibodies. Immunoprecipitations were performed with mild (M) or harsh (H) detergent concentrations. Harsh detergent conditions disrupt p62 incorporation into the dynein/dynactin complex. (G) p62-BioID-3×FLAG was immunoprecipitated from a stable HEK-293 cell line using α-FLAG antibodies and analyzed as described in (E) with α-FLAG and α-p150 antibodies. The signal intensities for p62-BioID-3×FLAG (magenta) and p150 (gray) are plotted as a fraction of the summed intensity of all fractions. The elution volumes of molecular weight standards are indicated (dashed lines).

The following figure supplement is available for figure 1:

**Figure supplement 1.** Schematic of the dynein/dynactin/activator complex.

To perform BioID experiments, we lysed cells in the presence of additional detergents (see Materials and methods), which disrupt both the dynein and dynactin complexes (*Figure 1D and F*). Disruption of the complexes makes it likely that our BioID experiments identified only proximal proteins that were modified with biotin prior to cell lysis. All BioID experiments with tagged dynein or dynactin subunits were performed in quadruplicate using a label-free quantitative proteomics approach to calculate the enrichment of each identified protein relative to a soluble BioID alone control (*Figure 2A*) (*Zhang et al., 2015*). 'Hits' were proteins with greater than 3-fold enrichment and p-values greater than 0.05 relative to the control. We first characterized the IC2 subunit of dynein, which is known to be centrally located within the tripartite dynein/dynactin/activator complex based on cryo-electron microscopy (cryo-EM) structural studies (*Figure 1A*) (*Chowdhury et al., 2015*; *Urnavicius et al., 2015*). Our IC2 BioID dataset identified all dynein subunits, as well as a number of dynactin subunits (*Figure 2B—D* and *Supplementary files 1* and *2*). In addition, the dataset contained the known activators BICD2, HOOK1, and HOOK3, as well as BICD1, a homolog of BICD2 that is a likely activator (*Figure 2D* and *Supplementary files 1* and *2*). The only known dynein activators that we did not identify were Spindly and RAB11FIP3. Spindly regulates mitotic-specific dynein functions in human cells (*Chan et al., 2009*; *Gassmann et al., 2010*), likely the reason we did not identify it in an unsynchronized cell population. RAB11FIP3 is poorly expressed in HEK-293T cells (*Huttlin et al., 2015*) (*Table 1*). These experiments show that the dynein IC is well positioned within the dynein/dynactin/activator complex for the BioID-based identification of activators.

To further explore the ability of BioID to report on the spatial organization of the dynein/dynactin/activator complex, we tagged additional dynein and dynactin subunits with BioID. Specifically, we generated five additional HEK-293 cell lines stably expressing BioID fused to the IC1, LIC1, LIC2, RB1, and TCTEX1 dynein subunits and analyzed these along with the tagged dynein IC2 and dynactin p62 subunits. Each BioID fusion protein incorporated into their respective complexes based on their ability to co-immunoprecipitate with dynein and dynactin (*Figure 3A and B*). Validating our approach, a protein-protein interaction network consisting of the hits shared between BioID-tagged subunits revealed that 13 of 20 hits present in three or more datasets were dynein/dynactin subunits and activators (*Figure 3—figure supplement 1* and *Supplementary file 3*). Further validation of our approach came from the presence of T-complex and prefoldin subunits as IC-specific hits (*Figure 3—figure supplement 1*), as prefoldin is a T-complex co-chaperone, and T-complex has been shown to interact with newly synthesized ICs (*Özdemir et al., 2016*; *Vainberg et al., 1998*).

Analyses of the dynein/dynactin hits enriched in each subunit dataset were also consistent with recent structural studies (*Figure 3C*) (*Chowdhury et al., 2015*; *Urnavicius et al., 2015*). IC1 and IC2 BioID samples detected more dynactin subunits than either LIC1 or 2, consistent with the dynein LIC being further away from dynactin compared to the IC (*Figure 3C*). With respect to activators, we found that the IC1, LIC1 and LIC2 dynein subunits and the p62 dynactin subunit identified dynein

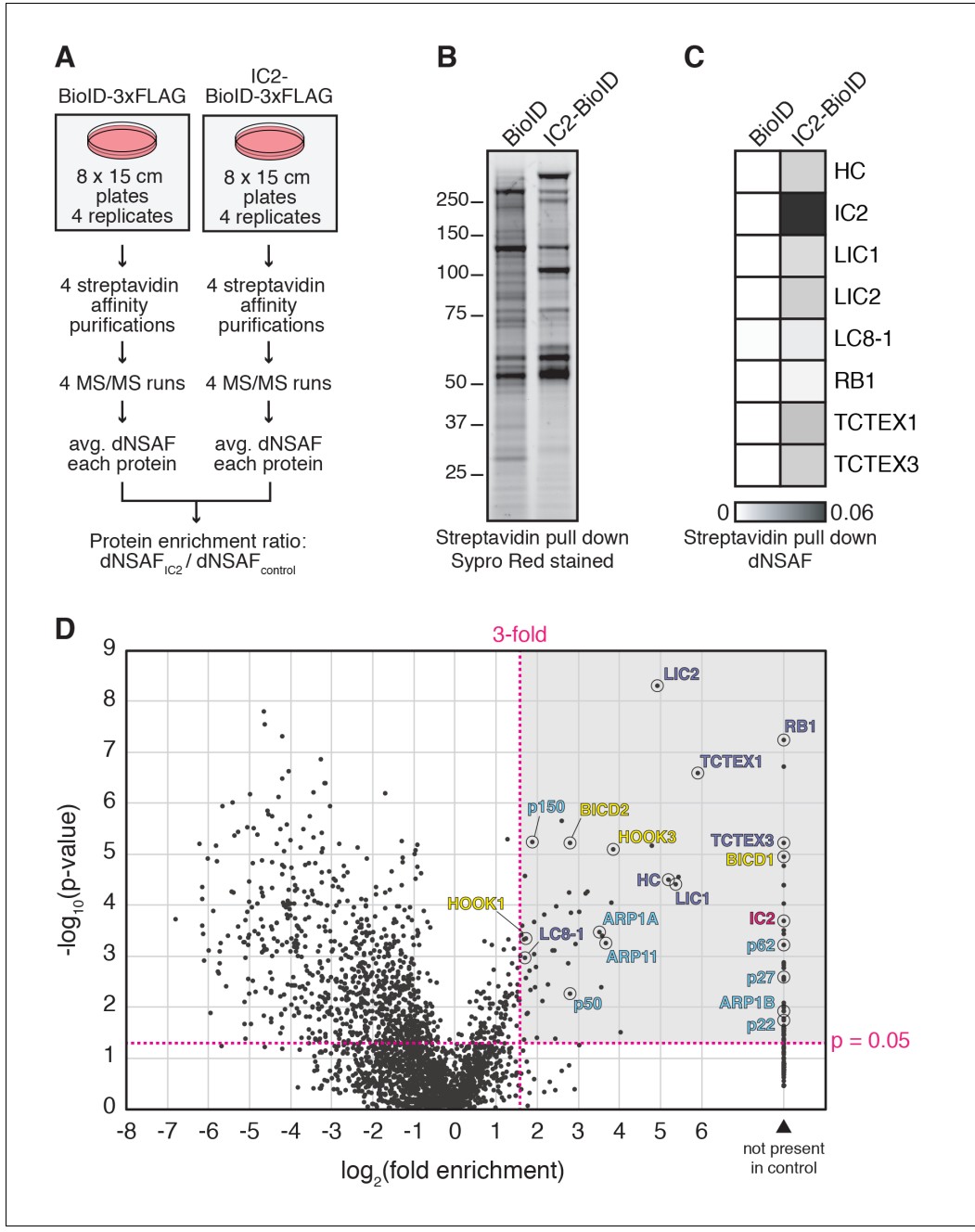

**Figure 2.** BioID with the dynein IC reports on activated dynein/dynactin/activator complexes in living human cells. (**A**) BioID experimental design. For each stably expressed BioID-tagged subunit reported in this study, quadruplicate samples were prepared, analyzed, and compared to a quadruplicate BioID only control. Fold enrichment was calculated as the ratio of dNSAF between the BioID-tagged subunit and the BioID control. (**B**) Biotinylated proteins were isolated from cells stably expressing either IC2-BioID or BioID by streptavidin affinity purification. A Sypro Red stained SDS-PAGE gel is shown. (**C**) MS/MS analysis of the immunoprecipitates from (**B**). Core dynein subunit dNSAF (*Zhang et al., 2015*) values are displayed as a heat map. (**D**) A volcano plot showing enrichment versus significance of proteins identified in IC2-BioID experiments relative to control (BioID alone) experiments. A quadrant (dashed magenta line) bounded by a p-value of 0.05 and 3-fold enrichment contained dynein (dark blue) and dynactin (light blue) subunits, as well as the known activators BICD2, HOOK1, and HOOK3, and the candidate activator BICD1 (yellow).

**Table 1.** Total protein expression levels of dynein and dynactin subunits, and activators or candidate activators or adaptors in HEK-293T cells. The number of peptides or phosphopeptides from HEK-293T cells (**Huttlin et al., 2015**) is shown. Activators or candidate activators identified in our secondary screen are highlighted in bold.

| Protein | Peptides | Phospho-peptides |
| --- | --- | --- |
| Dynein subunits | | |
| DYNC1H1 | 3395 | 9 |
| DYNC1I1 | 5 | 1 |
| DYNC1I2 | 281 | 76 |
| DYNC1LI1 | 499 | 179 |
| DYNC1LI2 | 130 | 29 |
| DYNLT1 | 44 | 0 |
| DYNLT3 | 15 | 0 |
| DYNLRB1 | 94 | 0 |
| DYNLRB2 | Not present | |
| DYNLL1 | 248 | 0 |
| DYNLL2 | 168 | 0 |
| Dynactin subunits | | |
| DCTN1 (p150) | 538 | 0 |
| DCTN2 (p50) | 338 | 13 |
| DCTN3 (p22) | 67 | 0 |
| DCTN4 (p62) | 152 | 0 |
| DCTN5 (p25) | 28 | 0 |
| DCTN6 (p27) | 29 | 0 |
| ACTR1A (Arp1) | 354 | 0 |
| ACTR1B (Arp1) | 222 | 0 |
| ACTR10 (Arp11) | 118 | 2 |
| CAPZA1 | 423 | 1 |
| CAPZA2 | 194 | 0 |
| CAPZB | 387 | 0 |
| Activators or candidate activators | | |
| **BICD1** | **35** | **0** |
| **BICD2** | **151** | **6** |
| BICDL1 | not present | |
| BICDL2 | not present | |
| **HOOK1** | **183** | **13** |
| HOOK2 | 57 | 2 |
| **HOOK3** | **130** | **0** |
| **CCDC88A (girdin)** | **247** | **51** |
| CCDC88B (gipie) | not present | |
| **CCDC88C (daple)** | **55** | **17** |
| SPDL1 (CCDC99) | 107 | 0 |
| RAB11FIP3 | 4 | 2 |
| **NIN** | **114** | **20** |
| **NINL** | **8** | **5** |
| TRAK1 | 17 | 8 |

*Table 1 continued on next page*

Redwine *et al.* eLife 2017;6:e28257. DOI: 10.7554/eLife.28257

*Table 1 continued*

| Protein | Peptides | Phospho-peptides |
|---------|----------|------------------|
| TRAK2 | 3 | 1 |
| HAP1 | not present | |
| RILP | not present | |

activators (*Figure 3C* and *Supplementary files 1* and *3*). This finding is consistent with the current structural model of the dynein/dynactin/activator complex (*Chowdhury et al., 2015*; *Urnavicius et al., 2015*). Thus, BioID provides spatial information about the large dynein complex, which is capable of moving in the cytoplasm. Importantly, these results also show that BioID experiments with the dynein IC and LIC subunits and dynactin p62 subunit can be used to identify activators, providing a method to discover dynein activators in other cell types or tissues.

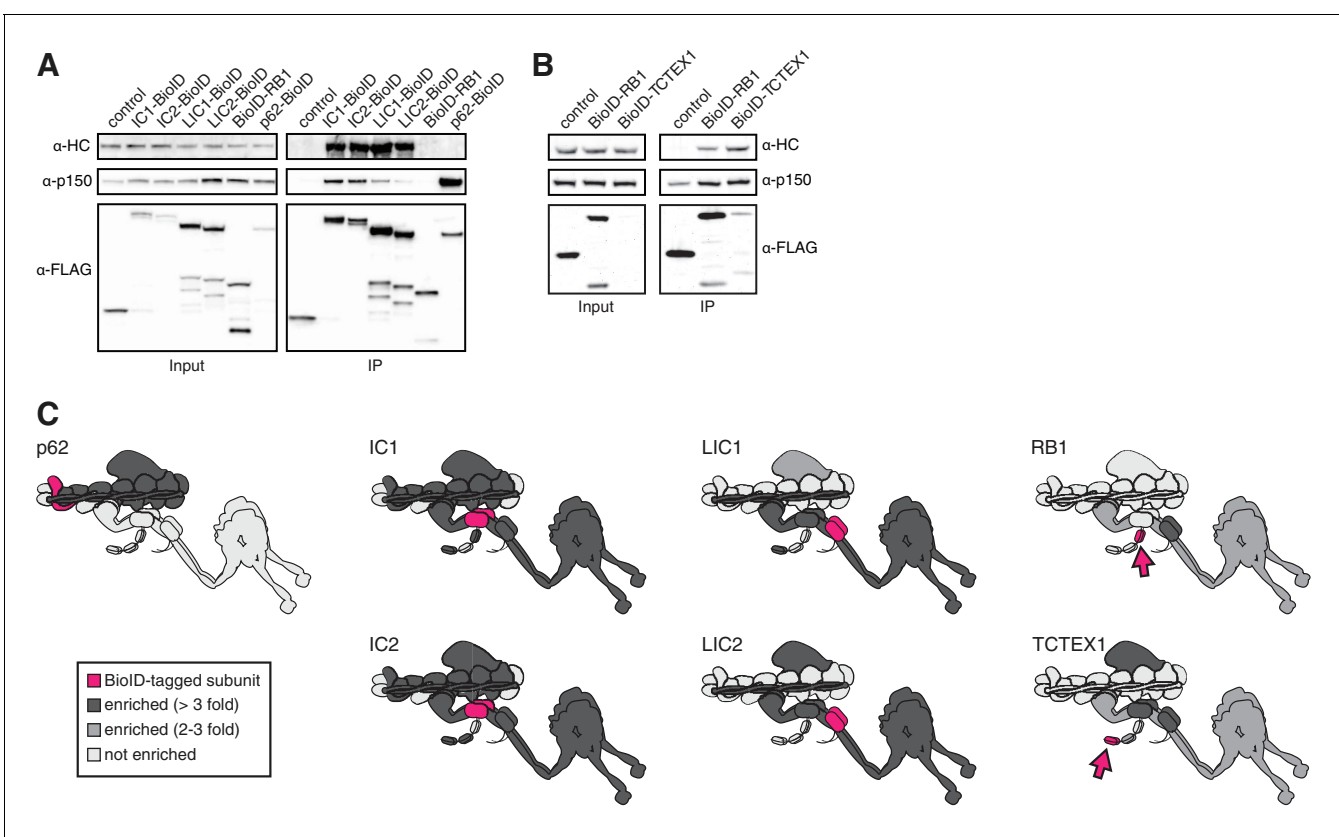

**Figure 3.** BioID reports on the spatial organization of the dynein/dynactin/activator complex. (**A and B**) Dynein (IC1, IC2, LIC1, LIC2, RB1, TCTEX1) and dynactin (p62) subunits tagged with BioID-3×FLAG were immunoprecipitated (for 16 hr in A or 2 hr in B) from stable HEK-293 cell lines using α-FLAG antibodies. All subunits incorporated into the dynein/dynactin complex based on Western blots with α-HC and α-p150 antibodies. (**C**) BioID experiments were performed with cells expressing the indicated dynein and dynactin subunits (magenta and magenta arrows). Other dynein and dynactin subunits enriched in the BioID experiments are shaded light gray (2–3 fold) or dark gray (≥3 fold), p<0.05 (Student's two-tailed t-test).

The following figure supplement is available for figure 3:

**Figure supplement 1.** Enriched and significant hits from dynein and dynactin BioID datasets were used to construct a protein-protein interaction network.

## Ninein and ninein-like constitute a new family of dynein activators

To identify novel dynein activators in the dynein/dynactin interactome, we performed a secondary screen (*Figure 4A*). Because all known activators contain long stretches of coiled coil (*Figure 4B*), we pooled datasets with known activators present (IC1, IC2, LIC1, LIC2, and p62) and selected a set of proteins with predicted coiled coils of at least 100 amino acids. We then expressed each predicted coiled coil domain tagged with GFP and 3×FLAG in HEK-293 cells (*Figure 4—figure supplement 1*). A hallmark of known activators is their ability to co-immunoprecipitate dynein and dynactin (*McKenney et al., 2014*; *Olenick et al., 2016*). Four proteins analyzed in our secondary screen, ninein (NIN), ninein-like (NINL), daple (CCDC88C) and girdin (CCDC88A), co-immunoprecipitated dynein and dynactin, as did our positive controls BICD2 and HOOK3 (*Figure 4C and D* and *Figure 4—figure supplement 1*). NIN was identified in the IC1 and IC2 BioID datasets, NINL and daple in the IC2 BioID dataset, and girdin in the LIC1 dataset (*Supplementary file 1*). Some construct optimization was necessary to determine the dynein/dynactin interacting region of each candidate activator and we used the literature to guide this process (*Casenghi et al., 2005*; *Schroeder and Vale, 2016*) (*Figure 4C and D* and *Figure 4—figure supplement 2A*).

To determine whether these candidate activators were part of activated dynein/dynactin complexes, we next performed in vitro single-molecule motility assays. We immunoprecipitated GFP-tagged BICD2 (aa 1–422), HOOK3 (aa 1–552), NIN (aa 1–693), NINL (aa 1–702), daple (aa 1–545) and girdin (aa 1–542) from HEK-293 cells and observed the motility of any co-purifying dynein and dynactin on microtubules using total internal reflection fluorescence (TIRF) microscopy. Our positive controls, BICD2 and HOOK3, exhibited robust processive motility, as did NIN and NINL (*Figure 4E–H*). In contrast, daple and girdin showed only a very modest ability to isolate activated dynein/dynactin complexes (*Figure 4—figure supplement 2B,C*). Because reconstituted purified dynein and dynactin occasionally show processive runs in the absence of an activator (*McKenney et al., 2014*; *Schlager et al., 2014*), we cannot yet conclude if daple and girdin are bona fide dynein activators.

The gold standard assay for dynein activators is to reconstitute dynein/dynactin/activator motility from purified components (*McKenney et al., 2014*; *Schlager et al., 2014*). To this end, we purified dynein and dynactin individually from HEK-293 cell lines stably expressing either IC2 or p62 tagged with SNAP or Halo tags (for fluorophore labeling) and 3×FLAG (for purification) (*Figure 5A* and *Figure 5—figure supplement 1A*). The coiled coil domains of BICD2, NIN, and NINL were tagged with GFP and purified from *E. coli* (*Figure 5—figure supplement 1A*). After reconstituting the complexes, we used near-simultaneous three-color TIRF microscopy to visualize the motility of single dynein/dynactin/activator complexes on microtubules. As expected, BICD2 activated and co-migrated with processively moving dynein and dynactin (*Figure 5B*). Both NIN and NINL also activated and co-migrated with moving dynein/dynactin complexes (*Figure 5C and D*). In the absence of an activator dynein/dynactin is largely stationary, with some diffusive and rare processive runs observed (*Figure 5E* and *Figure 5—figure supplement 1B*). Together, our results show that the BioID method can identify dynein activators, including the members of a new family of activators we discovered here: NIN and NINL.

## Identification of the interactomes of six dynein activators

A major goal in the transport field is to determine the molecular rules that govern cargo recognition and specificity. This is especially critical for dynein, which moves all microtubule minus-end-directed cargos in the cytoplasm. How many cargos does each activator recognize? Does each activator allow dynein/dynactin to recognize a specific subset of cargo, or is there overlap in the number of activators that can recognize any given cargo? To begin to address these questions, and to provide a starting point for future exploration of activator-cargo interactions, we used BioID to identify the interactomes of six dynein activators. We made both N- and C-terminal BioID fusions with each activator because, for the known activators BICD2 and HOOK3, their N-termini interact with dynein/dynactin and their C-termini bind to cargos or cargo adaptor proteins (*Cianfrocco et al., 2015*). We generated HEK-293 cell lines stably expressing full-length BICD1, BICD2, HOOK1, HOOK3, NIN, and NINL with BioID tags at either their N- or C-termini, and used MS/MS to detect activator proximal proteins that had been biotinylated in living cells from each cell line (*Figure 6A* and *Supplementary files 1* and *4*). As with our earlier experiments, we used detergent conditions (see Materials and methods) that made it likely we only identified proteins that were directly biotinylated

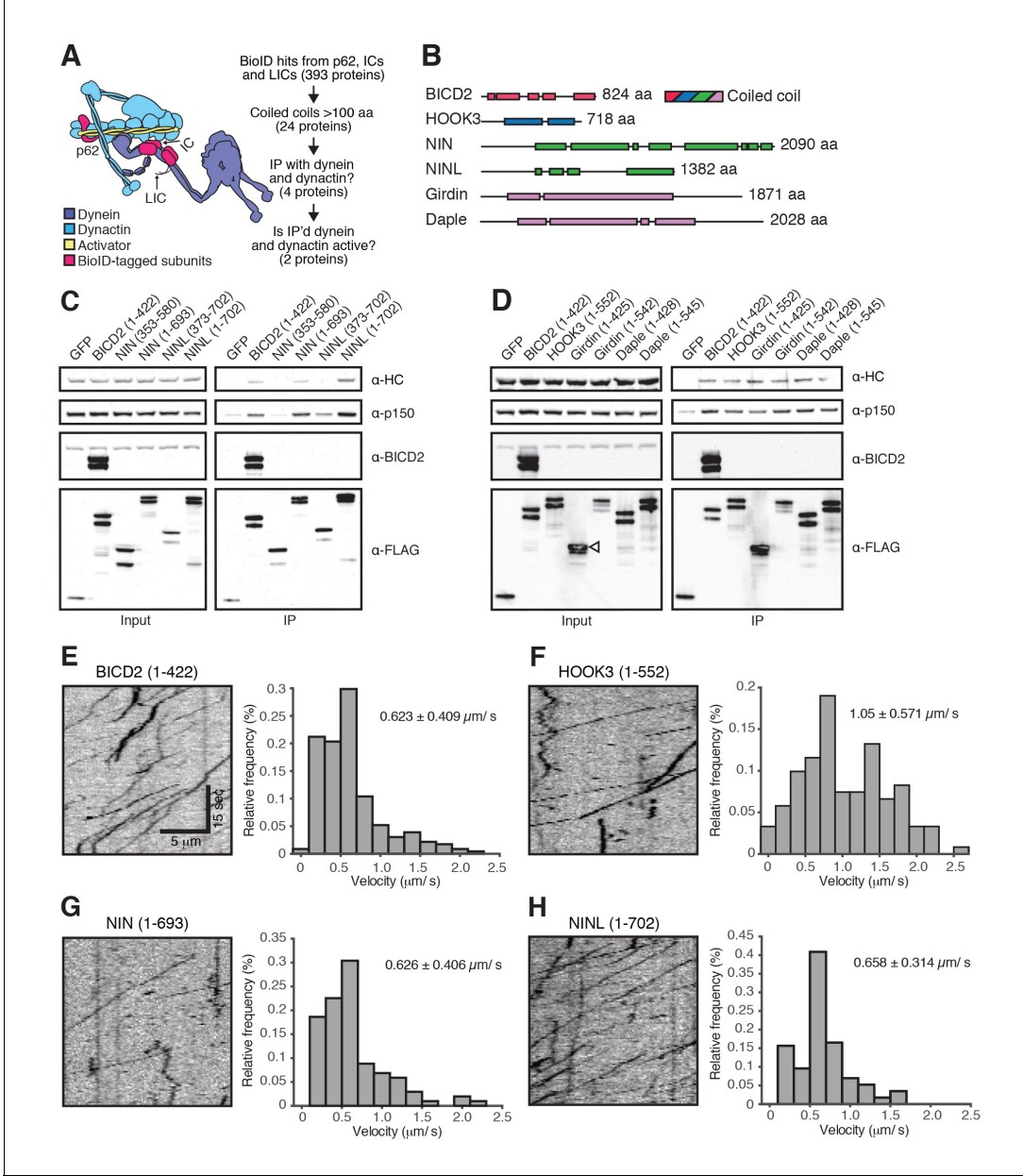

**Figure 4.** A secondary screen identifies candidate activators of dynein/dynactin motility. (**A**) A schematic of our secondary screen. (**B**) Location of predicted coiled coils (rectangles) in known and candidate dynein/dynactin activators. (**C, D**) Candidate and known (BICD2 and HOOK3) activators tagged with 3×FLAG were immunoprecipitated with α-FLAG antibodies from HEK-293 cells. Western blots with α-HC and α-p150 antibodies were used to determine which proteins co-immunoprecipitated dynein and dynactin. (**E—H**) The candidate NIN (1-693) and NINL (1-702) activators, as well as the known BICD2 (1–422) and HOOK3 (1–552) activators were tagged with GFP and 3×FLAG and were immunoprecipitated with α-FLAG antibodies from HEK-293 cells. The motility of immunoprecipitated dynein/dynactin/activator complexes was monitored by GFP fluorescence using TIRF microscopy. Kymographs (left) and velocity histograms (right) with mean velocity (± S.D.) shown, *n* is greater than 102. Data shown is analyzed from one technical replicate, although two technical replicates were collected for each activator and displayed similar trends.

The following figure supplements are available for figure 4:

**Figure supplement 1.** Candidate and known (BICD2 and HOOK3) activators were tagged with 3×FLAG and immunoprecipitated with α-FLAG antibodies from HEK-293 cells.

**Figure supplement 2.** (A) The amino acid sequences for HOOK1, HOOK2, HOOK3 and two HOOK-related proteins (daple and girdin) were downloaded from Uniprot and aligned using Clustal Omega.

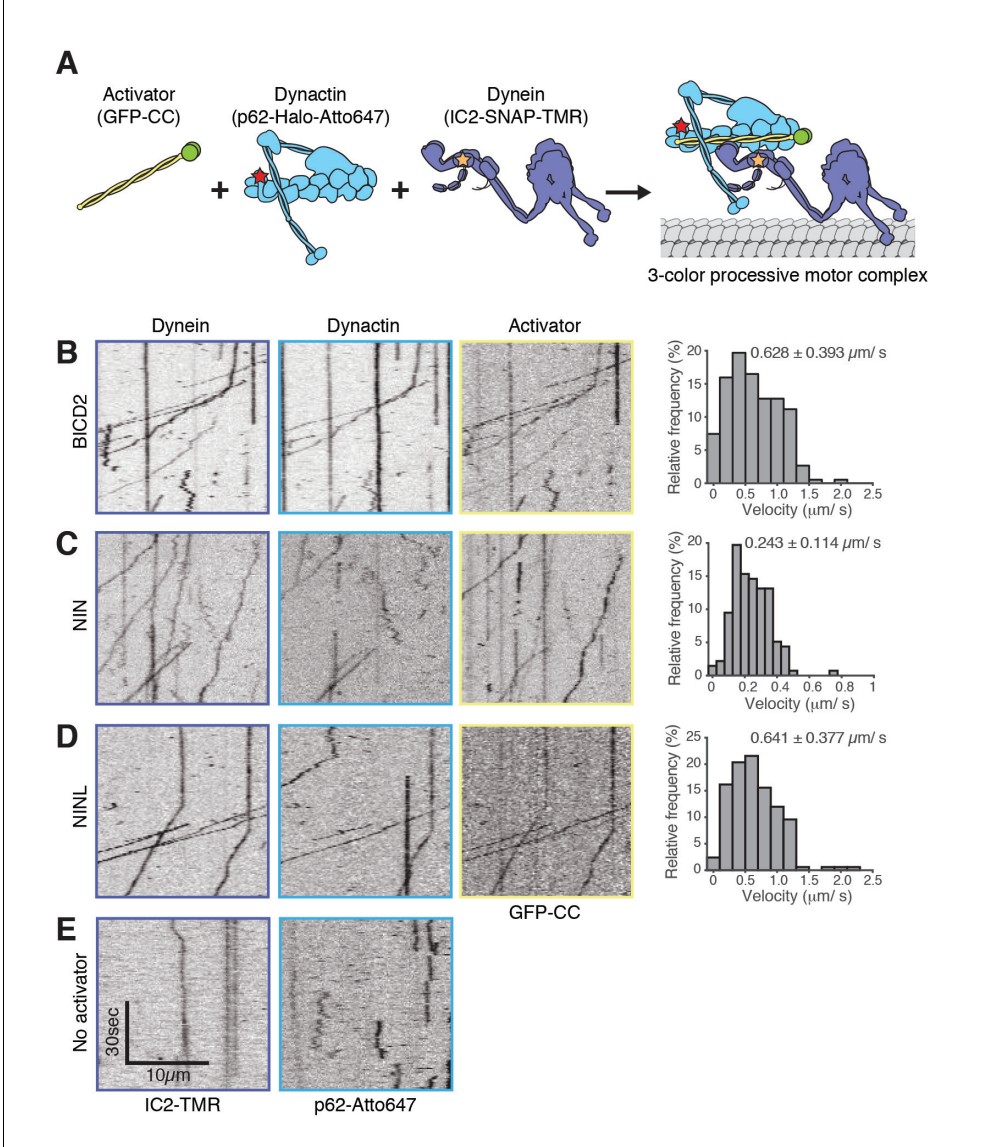

**Figure 5.** NIN and NINL are novel activators of dynein/dynactin motility. (**A**) A schematic of the components added to the single-molecule motility assay. Dynein (IC2-TMR; yellow star), dynactin (p62-Atto647; red star) and GFP-tagged (green spheres) activators (BICD2) or candidate activators (NIN and NINL) were purified separately, mixed, and the motility of the complex along microtubules was monitored by nearly simultaneous three-color TIRF microscopy. (**B-E** ).ymographs of each imaging channel (left) and velocity histograms (right) with mean velocity (± S. D.) are shown, *n* is greater than 146. NIN had a slower velocity in this assay compared to *Figure 4G*. This could be due to the lack of post-translational modifications in proteins expressed in *E. coli*; future work will be required to understand this. Data shown is analyzed from one technical replicate, although two technical replicates were collected for each activator and displayed similar trends.

The following figure supplement is available for figure 5:

**Figure supplement 1.** Purification of dynein, dynactin and activators.

and considered hits to be proteins with greater than 3-fold enrichment, p-values greater than 0.05 relative to a soluble BioID control, and average spectral counts greater than two.

BICD1 and 2, the best structurally characterized dynein activators, are known to be elongated structures (*Liu et al., 2013*; *Terawaki et al., 2015*; *Urnavicius et al., 2015*). Given this, and the likely

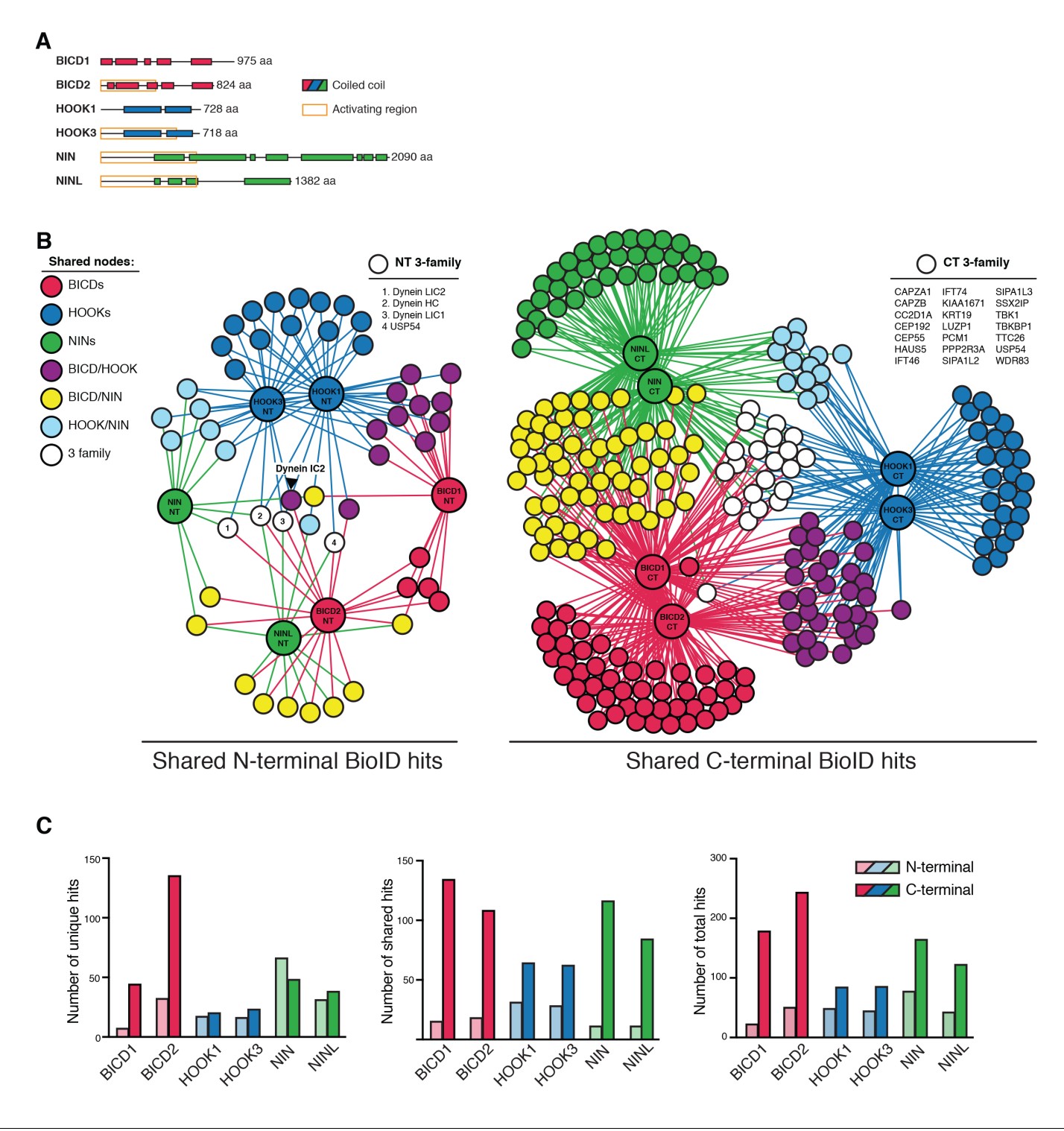

**Figure 6.** Dynein activators have distinct proteomes. (**A**) Location of predicted coiled coils in dynein activators, with minimal activating regions shown (orange rectangles). (**B**) Enriched and significant hits from N- and C-terminal datasets of six activators were used to construct two separate protein-protein interaction networks. Hits specific to an activator family (color-coded according to their respective activators), and hits shared between activator families (HOOK/BICD, purple; BICD/NIN, yellow; NIN/HOOK, cyan) are shown. White spheres ('3-family') represent hits enriched in at least one activator from each family. For this figure enrichment is ≥3 fold, significance is p<0.05, Student's two-tailed t-test; and average spectral counts are ≥2. The location of dynein and dynactin subunits and select hits discussed in the text are indicated. We note that we identified BICD1 in our BICD2 datasets and vice versa (**Supplementary files 1** and **4**). The same was true for HOOK1 and HOOK3, but not for NIN and NINL (**Supplementary files 1**

*Figure 6 continued on next page*

**Figure 6 continued**

and **4**). HOOK proteins have been shown to heterodimerize (**Xu et al., 2008**), whereas heterodimerization between BICD proteins has not been reported. (C) The number of total, unique (occurring in a single activator N- or C-terminal dataset), and shared (occurring in multiple activator N- or C-terminal datasets) hits for individual activator N- and C-termini are shown.

The following figure supplements are available for figure 6:

**Figure supplement 1.** An interaction map of the N- and C-terminal activator datasets combined.

**Figure supplement 2.** KIF1C is a novel HOOK3-interacting protein.

---

elongated nature of other activators, we reasoned that BioID would be ideally suited to report on the spatially separated functional differences between the N- and C-termini of activators. We analyzed the N- and C-terminal BioID datasets from all six activators. We first contrasted the N- and C-terminal datasets separately, in both cases seeking to reveal activator-specific interactions and those shared between multiple activators. The shared interactions were used to construct interaction networks revealing the connections between activators (**Figure 6B**). Overall, in comparison to N-terminal datasets, the C-terminal datasets had more unique hits (except for NIN), more hits shared between activators, and more total hits (**Figure 6C**).

We next identified proteins that were found in all activator families (BICD1/2, HOOK1/3, and NIN/NINL). When we performed this analysis for the N-terminal datasets only four out of 225 proteins were present in at least one dataset from each activator family (**Figure 6B**, white circles). Strikingly, three of these four proteins were dynein subunits (HC, LIC1, and LIC2), highlighting that the shared function associated with the N-termini of activators is their interaction with dynein. When we performed this analysis with the C-terminal datasets we found 21 out of 547 proteins were present in each activator family (**Figure 6B**). Gene ontology (GO) enrichment analysis (**Gene Ontology Consortium, 2015**) of these 21 proteins revealed an association with cellular locations that correlate with microtubule minus ends ('ciliary basal body', 'microtubule organizing center', and 'centrosome') (**Supplementary file 5**). Other GO terms that were enriched included 'F-actin capping complex', 'intraciliary transport particle B', and 'ciliary tip'. In contrast to the N-terminal dataset we did not detect any dynein subunits in the C-terminal dataset. Overall, this data highlights the power of the BioID technique to report on the distinct protein interactions of different regions of the dynein activators.

We next contrasted all 12 activator datasets (**Figure 6—figure supplement 1** and **Supplementary files 1** and **4**). This analysis allowed for the removal of hits shared between the N- and C-termini of activators and focus on C-terminal-specific activator hits, as this is the region of cargo interaction for characterized activators (**Cianfrocco et al., 2015**). Each activator had dozens of C-terminal specific hits (**Figure 6—figure supplement 1** and **Supplementary files 1** and **4**). These proteins are candidates for activator-specific cargos, cargo adaptors, or proteins that regulate how dynein connects to its cargo. GO term enrichment analysis of these activator-specific hits revealed several trends. The BICD2 C-terminal interactome was enriched for GO terms relating to cortical actin cytoskeleton structures, including adherens junctions and focal adhesions (**Supplementary file 5**), consistent with previous studies that have linked dynein to these cortical actin-based structures (**Ligon et al., 2001**; **Rosse et al., 2012**). The HOOK3 C-terminal interactome was enriched for GO terms relating to clathrin coated vesicles (**Supplementary file 5**). Both the NIN and NINL C-terminal interactomes were enriched for GO terms generally related to structures found at microtubule minus ends (**Supplementary file 5**).

We also analyzed the shared interactions of activators within the same family (e.g. BICD1/2, HOOK1/3 and NIN/NINL). Activators within families share significant sequence similarity (global identity, global similarity: BICD1/BICD2 = 53.6%, 65.2%; HOOK1/HOOK3 = 57.0%, 73.2%; NIN/NINL = 21.1%, 33.9%). Reflecting this protein conservation, activators from the same protein family had more C-terminal-specific overlap than activators from different families (**Table 2**). The BICD activator family was enriched for GO terms associated with the actin cytoskeleton. The HOOK family was enriched for GO terms related to kinesin motors (**Supplementary file 5**); we identified C-terminal-specific interactions of HOOK1 and HOOK3 with the kinesins KIF1C and KIF5B, as well as the

**Table 2.** Specific pairwise overlap between activator C-terminal BioID datasets. Twelve activator BioID datasets were contrasted (6 N-terminal and 6 C-terminal) to determine for each activator which of its C-terminal hits were specifically shared with other datasets. Those shared with N-terminal datasets were removed. The specific pairwise overlap of the remaining hits with each activator is reported (*n* and %). Input *n* = BICD1 (92), BICD2 (87), HOOK1 (39), HOOK3 (37), NIN (74), NINL (53). Only pairwise overlap is represented in this analysis; overlap with multiple activators (e.g. BICD1 overlap with both BICD2 and HOOK1) is not shown.

| | BICD1 | | BICD2 | | HOOK1 | | HOOK3 | | NIN | | NINL | |
|---|---|---|---|---|---|---|---|---|---|---|---|---|
| | *N* | % | *N* | % | *N* | % | *N* | % | *N* | % | *N* | % |
| BICD1 | | | 44 | 50.6 | 4 | 10.3 | 0 | 0 | 5 | 6.8 | 5 | 9.4 |
| BICD2 | 44 | 47.8 | | | 4 | 10.3 | 7 | 18.9 | 6 | 8.1 | 0 | 0 |
| HOOK1 | 4 | 4.3 | 4 | 4.6 | | | 14 | 37.8 | 3 | 4.1 | 1 | 1.9 |
| HOOK3 | 0 | 0 | 7 | 8.0 | 14 | 35.9 | | | 2 | 2.7 | 1 | 1.9 |
| NIN | 5 | 5.4 | 6 | 6.9 | 3 | 7.7 | 2 | 5.4 | | | 27 | 50.9 |
| NINL | 5 | 5.4 | 0 | 0 | 1 | 2.6 | 1 | 2.7 | 27 | 36.5 | | |

KIF5B associated light chains, KLC2 and KLC4 (*Supplementary file 4*). The NIN family was enriched for GO terms related to microtubule minus ends (*Supplementary file 5*).

Validating our approach, our analysis of activator- and activator family-specific hits identified several known activator-cargo interactions. HOOK1 and HOOK3 are members of the FHF complex, which is involved in endosomal sorting (*Xu et al., 2008*). We identified proteins in this complex (AKTIP/FTS and FAM160A2/FHIP), as well as two proteins that are homologous to FAM160A2 (FAM160A1, FAM160B1) as specifically enriched in the HOOK1 and HOOK3 datasets. FAM160A1 and FAM160A2 share sequence identity (global identity 36.4%, global similarity 50.5%), and FAM160A2 was reported to interact with HOOK3 and AKTIP/FTS in a recent high-throughput proteomics study (*Huttlin et al., 2015*). We also identified RIMBP3, a known HOOK1 interacting protein involved in spermatogenesis, as a HOOK1 C-terminal-specific hit (*Zhou et al., 2009*). In the BICD2 interactome, we identified RANBP2, a well-characterized BICD2-interacting protein that is responsible for targeting dynein/dynactin/BICD2 to nuclear pore complexes (*Splinter et al., 2010*). The NINL C-terminal interactome was specifically enriched for MICAL3, a protein that interacts with Rab8 and is localized to the base of primary cilia in a NINL-dependent manner (*Bachmann-Gagescu et al., 2015*).

We were intrigued by the presence of kinesin, including the kinesin-3 KIF1C, in the HOOK datasets (*Supplementary file 4*). Although Hook proteins are not known to interact with kinesins in humans, studies in filamentous fungi have linked dynein, Hook, and kinesin (*Bielska et al., 2014*). We used co-immunoprecipitation experiments to verify the interaction of HOOK3 with the dynein heavy chain, FAM160A2/FHIP, and KIF1C (*Figure 6—figure supplement 2*). Our BioID data had identified the interaction of HOOK3 with KIF1C and FAM160A2/FHIP as C-terminal-specific and the interaction with dynein/dynactin as N-terminal-specific. In agreement with our BioID data, immunoprecipitation experiments showed that the interaction of KIF1C and FAM160A2/FHIP with HOOK3 was specific to the C-terminus of HOOK3 (553–718) (*Figure 6—figure supplement 2*), while dynein's interaction was specific to the N-terminus of HOOK3 (aa 1–552) (*McKenney et al., 2014*). This data further validates the BioID approach and highlights how BioID can identify spatially restricted interactions. In addition, this data suggests that HOOK3 may represent a new class of dynein activator, one that not only activates dynein/dynactin, but can also recruit the opposite polarity motor KIF1C.

## Discussion

We applied proximity-dependent biotinylation to identify the dynein/dynactin and activator interactomes. Our data show that BioID reports on the spatial organization of both the tripartite dynein/dynactin/activator complex, as well as the domain organization of dynein activators. Using a secondary screen of our dynein/dynactin BioID data, we developed an approach to identify novel dynein activators. We identified ninein and ninein-like as a new family of dynein activators and two Hook-related proteins, girdin and daple, as candidate dynein/dynactin activators. Our analysis of the

activator interactomes suggests that there are dozens of unique interactions for each activator, as well as shared interactions particularly among activators of the same family. We propose that these proteins represent novel dynein cargos, cargo adaptors or regulators of motor-cargo interactions and that each activator will link dynein to multiple cargos.

## BioID provides spatial information about the large cytosolic dynein transport machinery

We tagged seven distinct dynein or dynactin subunits with the BioID tag. Our analysis of their interactomes suggests that BioID can provide spatial information about the large dynein machinery, capable of moving in the live cells we used for our experiments. The recent cryo-EM structures of the dynein/dynactin/activator complex (*Chowdhury et al., 2015*; *Urnavicius et al., 2015*) allowed us to roughly map the interactions we identified. The interactomes of each dynein or dynactin subunit identified other dynein and dynactin subunits that were located in close proximity based on these structural studies. Important for future discovery efforts, we found that proteins that were in the vicinity of the activator (the dynein ICs and LICs and the p62 subunit of dynactin) had interactomes containing activators. Thus, tagging these proteins with proximity-dependent biotinylation tags will allow future efforts to identify dynein activators in other cell types and tissues.

We also found that each activator, all of which contain long stretches of predicted coiled coil and likely have elongated structures, had largely non-overlapping protein interactions depending on whether their N- or C-terminus was tagged with BioID. Our data agrees with published data showing that the N-termini of activators interact with dynein/dynactin and that the C-termini with cargos or cargo adaptors (*Cianfrocco et al., 2015*). We also identified a novel interaction between the dynein activator HOOK3 and the kinesin-3 KIF1C. Both our BioID data and confirmatory immunoprecipitation experiments showed that dynein interacts with the N-terminus of HOOK3 (*McKenney et al., 2014*), while kinesin interacts with its C-terminus. This further highlights the ability of BioID to provide spatial information about protein-protein interactions and also raises the exciting possibility that HOOK3 is a scaffold for plus- and minus-end-directed microtubule-based motors.

## BioID identifies ninein and ninein-like as members of a new family of dynein/dynactin activators

Our secondary screen of predicted coiled coil-containing proteins found in the dynein/dynactin interactome identified four proteins that could co-immunoprecipitate with dynein and dynactin: NIN, NINL, girdin and daple. Further analysis of these proteins demonstrated that NIN and NINL activated dynein/dynactin motility in single-molecule motility assays, while girdin and daple did not. Here, we focused on determining if predicted coiled coil-containing proteins could activate dynein/dynactin motility. Future analysis of the hits found in the dynein/dynactin interactome could identify additional positive or negative regulators of dynein motor activity. It is possible that there will be dynein activators lacking large stretches of coiled coil, which were not assessed in our secondary screen.

Both NIN and NINL localize to the centrosome, are involved in microtubule nucleation, and have been shown previously to immunoprecipitate with dynein/dynactin (*Casenghi et al., 2005*, *2003*; *Delgehyr et al., 2005*; *Wang et al., 2015*). The ability of NIN and NINL to activate dynein suggests that they control their own, as well as any associated proteins, recruitment to the centrosome or other sites of microtubule nucleation. NINL has also been implicated in dynein-based vesicle trafficking (*Dona et al., 2015*), in support of the idea that each activator has multiple cargos (see below).

Girdin and daple are Hook-related proteins (*Simpson et al., 2005*), which also act as guanine nucleotide exchange factors for small G proteins (*Aznar et al., 2016*, *2015*). While both robustly co-immunoprecipitated dynein and dynactin, they did not conclusively activate dynein/dynactin motility. We used girdin and daple constructs that were identical in length to a HOOK3 construct that could activate motility. However, it is possible that longer girdin or daple constructs will be required for activation or that post-translational modifications regulate their ability to activate dynein/dynactin motility. For example, both daple and girdin are phosphoproteins (*Table 1*) (*Huttlin et al., 2015*). It is also possible that girdin and daple regulate dynein by blocking the ability of motility-inducing activators to bind to dynein. CCDC88B/gipie is related to girdin and daple and interacts with dynein

and dynactin (*Ham et al., 2015*). We did not identify gipie in our screen, likely because it is not expressed in HEK-293T cells (*Table 1*) (*Huttlin et al., 2015*). Future studies of these Hook-related proteins will be aimed at exploring if and how they regulate dynein/dynactin activity.

## Activators have many new candidate cargos

Our findings suggest that the number of dynein activators is much smaller than the number of dynein cargos, strongly implying that each activator links dynein to multiple cargos. There are hints of this concept in the literature as BICD2 interacts with both Rab6 and RanBP2 (*Hoogenraad and Akhmanova, 2016*), and in *Drosophila* the RNA binding protein egalitarian (*Dienstbier et al., 2009*). Consistent with this, our GO analysis of the interactomes of six distinct dynein activators suggests that each of these activators is involved in multiple dynein-based functions. Overall, our data imply a tiered mode of dynein regulation in which activators, such as members of the BICD, HOOK and NIN families constitute the first step in cargo recognition, but additional layers must be required to achieve cargo specificity.

Our data raise a number of interesting questions for future exploration. How are activators released from dynein/dynactin? Which factors mediate this? Given that the dynein/dynactin machinery may be relatively invariant compared to activators, are activators exchanged? And, if so which factors mediate this? Are there proteins that bind to the same region of dynein/dynactin as activators, but don't activate motility? Finally, are activators promiscuous and if so what is the balance of stochastic versus regulated motor-cargo interactions? Our dynein transport machinery interactome provides a rich dataset to address these fundamental questions.

# Materials and methods

## Molecular cloning and generation of stable cell lines

All plasmids used in this study were constructed by PCR and Gibson isothermal assembly. BioID G2 (*Kim et al., 2016a*) was the kind gift of Kyle Roux (Sanford School of Medicine, University of South Dakota). ORFs (isoforms indicated where applicable) were obtained from several sources. IC1 (isoform 2, 628 aa), IC2 (isoform 2C, 612 aa), LIC1, Roadblock (isoform 1, 96 aa), TCTEX1, and p62 (isoform 1, 460 aa) were amplified from a human RPE1 cell cDNA library (generated in the Reck-Peterson lab). LIC2 (isoform 1, 492 aa) and HOOK1 (isoform 1, 728 aa) were obtained from the Harvard Medical School PlasmID Repository, BICD2 (isoform 1, 824 aa) from Thermo Fisher Scientific (Waltham, MA), BICD1 (isoform 1, 975 aa) from Genescript (Piscataway, NJ), HOOK3 from GE Dharmacon (Lafayette, CO), and NIN was the kind gift of Dr. Yi-Ren Hong (Department of Biochemistry, Kaohsiung Medical University, Taiwan). NINL (isoform 1, 1382 aa) was synthesized in segments (IDT; Coralville, IA) and assembled by Gibson isothermal assembly. For constitutive expression, ORFs were inserted into pcDNA5/FRT (Invitrogen). IC1, IC2, LIC1, LIC2, and p62 were constructed as C-terminal fusions with BioID (e.g. pcDNA5-FRT-IC1-5×GA-BioID-3×FLAG); Roadblock1 and TCTEX1 were constructed as N-terminal fusions (e.g. pcDNA5-FRT-BioID-5×GA-Roadblock1-3×FLAG); and activators (BICD1, BICD2, HOOK1, HOOK3) were constructed as both N- and C-terminal fusions. To obtain inducible expression, NIN-BioID, BioID-NIN, NINL-BioID, BioID-NINL, and a BioID control were inserted into pcDNA5/FRT/TO (Invitrogen/ Thermo Fisher Scientific). All constructs had 5×glycine-alanine linkers added between BioID and the ORFs to provide flexibility between the modules. All constructs were sequence verified and expression was verified by Western blotting with an anti-FLAG M2-HRP antibody (Sigma-Aldrich; Saint Louis, MO).

For all experiments, stable and transiently transfected cell lines were maintained at 37°C with 5% $CO_2$ in Dulbecco's Modified Eagle Medium (DMEM, Corning; Tewksbury, MA) supplemented with 10% fetal bovine serum (FBS, Gibco/ Thermo Fisher Scientific) and 1% Penicillin/Streptomycin (PenStrep, Corning). HEK293-T cells (purchased from ATCC, Manassas, VA, catalog number: CRL-3216) were used for all transient transfections. Flp-In™ T-REx™293 cells (purchased from Thermo Fisher, catalog number: R78007), which constitutively express the Tet repressor, were used to generate the stable cell lines. We confirmed that both cell lines were free of mycoplasma contamination. Stable cell lines were generated by transfection with Lipofectamine 2000 (Thermo Fisher Scientific) and a combination of the appropriate pcDNA5 construct and pOG44, which expresses Flipase. After recovery from transfection, cells were grown in DMEM containing 10% FBS, 1% PenStrep, and 50

µg/mL Hygromycin B. Colonies were isolated, expanded, and screened for expression of the fusion proteins by Western Blotting with an anti-FLAG M2-HRP antibody (A8592, Sigma-Aldrich; see 'Western analysis and antibodies' below for details).

### Protein sequence analysis

Protein sequences were downloaded from UniProt (*The UniProt Consortium, 2017*). Multiple sequence alignments were calculated with Clustal Omega, and pairwise alignments used to calculate percent identity and similarity between proteins were calculated with EMBOSS Needle (*Li et al., 2015*).

### BioID: cell growth and sample preparation

To initiate a BioID experiment, low passage cells were plated at 20% confluence in 15 cm dishes as four replicates, with each replicate consisting of $8 \times 15$ cm plates. After 24 hr, biotin was added to the media to a final concentration of 50 µM, and the cells were incubated for an additional 16 hr. Tetracycline was added to tetracycline-inducible stable cell lines (1 µg/mL final concentration) at the same time as biotin. After decanting the media, cells were dislodged from each plate by pipetting with ice-cold PBS. Cells were centrifuged at 1000 x g for 2 min and the PBS was decanted. Cells were washed once more with ice cold PBS before proceeding to cell lysis. Cells were resuspended and lysed in 10 mL RIPA buffer (50 mM Tris-HCl, pH 8.0; 150 mM NaCl, 1% (v/v) NP-40, 0.5% (w/v) sodium deoxycholate, 0.1% (w/v) SDS, 1 mM DTT, and protease inhibitors (cOmplete Protease Inhibitor Cocktail, Roche; Switzerland) by gentle rocking for 5–10 min at 4°C. The cell lysate was clarified via centrifugation at 66,000 x g for 30 min in a Type 70 Ti rotor (Beckman Coulter; Brea, CA) at 4°C. The clarified lysate was retrieved and dialyzed twice against dialysis buffer (50 mM Tris-HCl, pH 8.0; 150 mM NaCl, 1 mM DTT, 0.01% Triton X-100) for 2 hr per exchange. The dialysate was retrieved, supplemented with fresh protease inhibitors, and combined with 1 mL streptavidin-conjugated beads (Dynabeads MyOne Streptavidin T1, Thermo Fisher Scientific) and incubated overnight at 4°C with gentle rocking. Bead/lysate mixtures were collected on a magnetic stand into a single 2 mL round-bottom microcentrifuge tube. The beads were then washed 4 times with 2 mL RIPA buffer, with immobilization and solution removal performed on a magnetic stand. To elute bound immobilized proteins, the beads were boiled for 10 min at 100°C in 100 µL elution buffer (50 mM Tris, pH 6.8, 2% SDS (w/v), 20 mM DTT, 12.5 mM EDTA, 2 mM biotin). Typically, 10 µL was analyzed by SDS-PAGE and Sypro Red staining and the remaining eluate (90 µL) was diluted to a final volume of 400 µL with 100 mM Tris-HCl, pH 8.5. 100% trichloroacetic acid was added to a final concentration of 20% and the solution was incubated overnight at 4°C. The precipitate was collected by centrifugation at maximum speed in a microcentrifuge for 30 min at 4°C. The supernatant was removed, the pellet was washed with 500 µL ice cold 100% acetone, and was centrifuged at maximum speed in a microcentrifuge for 10 min at 4°C. The acetone was removed, and the wash was repeated. After removing the final acetone wash, the pellet was dried in a laminar flow cabinet for 30–60 min.

### Mass spectrometry

#### Preparation of peptide mixtures

TCA-precipitated protein samples from streptavidin affinity purifications or FLAG immunoprecipitations were solubilized in 30 µl of freshly made 0.1 M Tris-HCl, pH 8.5, 8 M urea, 5 mM TCEP (Tris [2-Carboxylethyl]-Phosphine Hydrochloride, Pierce/ Thermo Fisher Scientific). After 30 min at room temperature, freshly made 0.5 M 2-Chloroacetamide (Sigma-Aldrich) was added to a final concentration of 10 mM, and the samples were left at room temperature for another 30 min in the dark. Endoproteinase Lys-C (Roche) was first added at an estimated 1:100 (wt/wt) enzyme to protein ratio, for at least 6 hr at 37°C. Urea was then diluted to 2 M with 0.1 M Tris-HCl, pH 8.5, $CaCl_2$ was added to 0.5 mM, and modified trypsin (Promega; Madison, WI), 1:100 (wt/wt), was added for over 12 hr at 37°C. All enzymatic digestions were quenched by the addition of formic acid to 5% final concentration.

#### Data acquisition

Each trypsin-digested sample was analyzed independently by Multidimensional Protein Identification Technology (MudPIT) as described previously (*Washburn et al., 2001*; *Wolters et al., 2001*).

Peptide mixtures were pressure-loaded onto a 250 μm fused-silica column packed first with 2 cm of 5 μm C18 reverse phase particles (Aqua, Phenomenex; Torrance, CA), followed by 3 cm of 5 μm strong cation exchange material (Partisphere SCX, Whatman/ GE Healthcare Life Sciences; Pittsburg, PA). The loaded microcapillary columns were then connected to a 100 μm fused-silica column pulled to a 5 μm tip using a P 2000 CO2 laser puller (Sutter Instruments) packed with 8 cm of 5 μm C18 reverse phase particles. Loaded and assembled microcapillaries were placed in line with either a LTQ ion trap mass spectrometer (Thermo Fisher Scientific; for all datasets except *Figure 2D*) or a Velos Orbitrap Elite mass spectrometer (Thermo Fisher Scientific; for *Figure 2D*), both of which were interfaced with quaternary Agilent 1100 quaternary pumps (Agilent Technologies). Overflow tubing was used to decrease the flow rate from 0.1 mL/min to about 200–300 nL/min. During the course of fully automated chromatography, ten 120 min cycles of increasing salt concentrations followed by organic gradients slowly released peptides directly into the mass spectrometer (*Florens and Washburn, 2006*). Three different elution buffers were used: 5% acetonitrile, 0.1% formic acid (Buffer A); 80% acetonitrile, 0.1% formic acid (Buffer B); and 0.5 M ammonium acetate, 5% acetonitrile, 0.1% formic acid (Buffer C). The last two chromatography steps consisted in a high salt wash with 100% Buffer C followed by the acetonitrile gradient. The application of a 2.5 kV distal voltage electrosprayed the eluting peptides directly into LTQ linear ion trap mass spectrometers equipped with a nano-LC electrospray ionization source (ThermoFinnigan). For LTQ MS runs, each full MS scan (from 400 to 1600 m/z) was followed by five MS/MS events using data-dependent acquisition where the first most intense ion was isolated and fragmented by collision-induced dissociation (at 35% collision energy), followed by the 2nd to 5th most intense ions. For Orbitrap Elite MS runs, full MS spectra were recorded on the peptides over a 400 to 1600 m/z range, followed by 10 tandem mass (MS/MS) events sequentially generated in a data-dependent manner on the first to tenth most intense ions selected from the full MS spectrum (at 35% collision energy). Dynamic exclusion was enabled for 90 s.

## Data analysis

RAW files were extracted into ms2 file format (*McDonald et al., 2004*) using RAW_Xtract (J.R. Yates, Scripps Research Institute). MS/MS spectra were queried for peptide sequence information on a 157-node dual processor Beowulf Linux cluster dedicated to SEQUEST analyses (*Eng et al., 1994*). MS/MS spectra were searched without specifying differential modifications against a protein database consisting of 55508 human proteins (downloaded from NCBI on 2014-02-04), and 177 usual contaminants (such as human keratins, IgGs, and proteolytic enzymes). In addition, to estimate false discovery rates, each non-redundant protein entry was randomized. The resulting 'shuffled' sequences were added to the database and searched at the same time as the 'forward' sequences. To account for carboxamidomethylation by CAM, +57 Da were added statically to cysteine residues for all the searches.

Results from different runs were compared and merged using CONTRAST (*Tabb et al., 2002*). Spectrum/peptide matches were only retained if peptides were at least seven amino acids long and fully tryptic. The DeltCn had to be at least 0.08, with minimum XCorrs of 1.8 for singly-, 2.5 for doubly-, and 3.5 for triply-charged spectra, and a maximum Sp rank of 10. Finally, combining all runs, proteins had to be detected by at least two such peptides, or 1 peptide with two independent spectra. Proteins that were a subset of others were removed.

NSAF7 (Tim Wen) was used to create the final report on all detected proteins across the different runs, calculate their respective distributed Normalized Spectral Abundance Factor (dNSAF) values, and estimate false discovery rates (FDR).

Spectral FDR is calculated as:

$$\text{FDR} = \frac{2 \times \text{Shuffled Spectral Counts}}{\text{Total Spectral Counts}} \times 100$$

Protein level FDR is calculated as:

$$\text{Protein FDR} = \frac{\text{Shuffled Protein}}{\text{Total Protein}} \times 100$$

Under these criteria the overall FDRs at the spectra and peptide levels were less than 1%.

To estimate relative protein levels, dNSAFs were calculated for each non-redundant protein, as described (*Florens et al., 2006*; *Mosley et al., 2009*; *Paoletti et al., 2006*; *Zhang et al., 2010*; *Zybailov et al., 2006*). Average dNSAFs were calculated for each protein using replicates with non-zero dNSAF values. Selected average dNSAF values for proteins detected in FLAG immunoprecipitations and streptavidin affinity purifications (*Figures 1C* and *2C*) were visualized using Multi Experiment Viewer. Enrichment of proteins in streptavidin affinity purifications from BioID-tagged stable cell lines relative to a control BioID stable cell line were calculated as the ratio of average dNSAF (ratio = avg. $dNSAF_{ORF-BioID}$: avg. $dNSAF_{BioID}$). The volcano plot in *Figure 2D* was generated by plotting the $\log_2$(fold enrichment) against the $-\log_{10}$(p-value), where the p-value (2-tailed Student's T-test) was generated by comparing the replicate dNSAF values of IC2-BioID to the BioID control. Mapping of dynein/dynactin subunits detected in dynein core subunit BioID datasets was performed by first calculating fold enrichment and p-values. Hits within the dynein/dynactin complex were mapped if they had either 2–3 fold or >3 fold enrichment and had p-values<0.05 (*Figure 3C*).

To compare and contrast BioID datasets (core [*Figure 3—figure supplement 1*], activator NT [*Figure 6B*], activator CT [*Figure 6B*], and combined activator NT-CT [*Figure 6—figure supplement 1*]) we first sorted proteins based on enrichment and significance. Protein hits with >3 fold enrichment, p-values<0.05 (Students two-tailed t-test), and average spectral counts > 2 were used for all of these analyses. One replicate each for NIN-BioID and HOOK3-BioID were discarded due to extremely low overall spectral counts, and one replicate for BioID-BICD1 was lost due to a faulty LC column. Hits were contrasted to generate categorized lists of hits (http://bioinformatics.psb.ugent.be/webtools/Venn/). For network analysis, we constructed protein-protein interaction networks using Cytoscape (*Figure 3—figure supplement 1*, *Figure 6B*, *Figure 6—figure supplement 1*; cytoscape.org).

For the core dynein/dynactin subunit network we contrasted seven datasets (IC1-BioID, IC2-BioID, LIC1-BioID, LIC2-BioID, BioID-TCTEX1, BioID-RB1, and p62-BioID) to determine the proteins unique to each dataset and those shared between multiple datasets. Those shared between datasets were used to construct the network (*Figure 3—figure supplement 1*).

Similarly, we constructed networks comprising the 6 N-terminal activator datasets (NT: BioID-BICD1, BioID-BICD2, BioID-HOOK1, BioID-HOOK3, BioID-NIN, BioID NINL), 6 C-terminal activator datasets (CT: BICD1-BioID, BICD2-BioID, HOOK1-BioID, HOOK3-BioID, NIN-BioID, NINL-BioID) (*Figure 6B* and *Supplementary file 4* ['NT hits' tab and 'CT hits' tab]), and a combination of all 12 NT and CT datasets (*Figure 6—figure supplement 1* and *Supplementary file 4* ['NT-CT combined hits' tab]). Activator NT-specific hits (*Supplementary file 4* ['NT hits' tab, 'NT Specific' columns]) were tabulated. The remaining shared hits (*Supplementary file 4* ['NT hits' tab, 'NT Shared' columns]) were assigned their respective activator interactions and then used to create a network. The same process was repeated to construct a CT-network. Activator CT-specific hits (*Supplementary file 4* ['CT hits' tab, 'CT Specific' columns]) were tabulated. The remaining shared hits (*Supplementary file 4* ['CT hits' tab, 'CT Shared' columns]) were assigned their respective activator interactions and then used to create a network. To construct the complete NT-CT network all 12 activator datasets were contrasted to determine the unique and shared hits for each activator NT and CT (*Supplementary file 4* ['NT-CT hits' tab, 'NT Specific', 'CT Specific', 'NT Shared' and 'CT Shared' columns]). Shared hits were used to construct the 12-way network. Hits used for network analysis refer to proteins with designated NCBI gene names. Predicted proteins lacking NCBI gene names were omitted from this analysis.

All networks were organized using the Cytoscape 'yFiles Layouts-Organic' option. Regions of the interaction map were color coded and then selected for gene ontology analysis using the 'cellular component' option (GO, geneontology.org) (*Gene Ontology Consortium, 2015*). The GO terms with p-values<0.05 were tabulated (*Supplementary file 5*).

## Immunoprecipitations

For small-scale immunoprecipitations (*Figures 1D, F*, *3A and B*) from stable cell lines, cells were split into 10 cm dishes at 40% confluence the day before harvesting. For immunoprecipitations from transiently transfected cells (*Figure 4C and D* and *Figure 4—figure supplement 1A*, *Figure 6—figure supplement 2*), cells were plated on 10 cm dishes at 10–15% confluence the day before transfection. At least two technical replicates were performed for the experiments in *Figures 1*, *3* and *4*. Transfections were performed with Lipofectamine 2000 (Invitrogen) and 2–4 μg (kept constant within an

experiment) of transfection-grade DNA (Purelink midi prep kit, Invitrogen/ Thermo Fisher Scientific) per dish in OPTI-MEM media. After 6 hr the media was exchanged for DMEM containing 10% FBS and 1% PenStrep. Cells were then grown for 24–48 hr before lysate preparation. For both approaches (transient or stable cell lines), cell collection, lysis, and immunoprecipitation conditions were the same. Cells were collected by decanting the media, and washing the cells off the dish with ice-cold PBS. Cells were collected by centrifugation at 1000 x g for 2 min, washed again with PBS, and then transferred with PBS to Eppendorf tubes for lysis. Cells were centrifuged at 2000 rpm in a microcentrifuge for 1 min and the supernatant was removed. For *Figure 1D and 1F*, cells were lysed in 500 µL of either 'mild' detergent lysis buffer (50 mM Tris-HCl, pH 7.4, 100 mM NaCl, 0.2% Triton X-100, 1 mM DTT, 0.5 mM ATP, 1X protease inhibitor cocktail (cOmplete, Roche)) or 'harsh' detergent lysis buffer (50 mM Tris-HCl, pH 8.0, 150 mM NaCl, 0.1% SDS (wt/v), 0.5% sodium deoxycholate (wt/v), 1% NP40 (v/v), 0.5 mM ATP, 1 mM DTT, 1X protease inhibitor cocktail). For immunoprecipitions of dynein and dynactin subunits from stable cell lines (*Figure 3*), immunoprecipitations followed by single-molecule motility (*Figure 4* and *Figure 4—figure supplement 2*) and those used to assess coiled coil protein binding (*Figure 4—figure supplement 1*), cells were lysed in 500 µL dynein lysis buffer (25 mM HEPES, pH 7.4, 50 mM potassium acetate, 2 mM magnesium acetate, 1 mM EGTA, 10% glycerol, and 1 mM DTT) supplemented with 0.2% Triton X-100, 0.5 mM Mg-ATP, and 1X protease inhibitors (cOmplete Protease Inhibitor Cocktail, Roche). For all samples, lysis was accomplished with gentle mixing at 4°C for 20 min. All lysates were centrifuged at maximum speed in a 4°C microcentrifuge for 15 min. The clarified lysate was retrieved and added to 50 µL packed volume of anti-FLAG M2 agarose (Sigma-Aldrich) and incubated for either 2 (*Figures 3B* and *4C and D* and *Figure 4—figure supplement 1A*, *Figure 6—figure supplement 2*) or 16 hr (*Figures 1D, F* and *3A*) at 4°C. Cells were washed four times in the same buffer they were lysed in and elutions were performed with 50 µL of each lysis buffer supplemented with 0.4 mg/mL 3×FLAG peptide.

One large scale FLAG-immunoprecipitation experiment was carried out (*Figure 1B and C*) in order to analyze the composition of dynein complexes isolated from cells expressing IC2-BioID-3×FLAG (*Figure 1B and C*). Four replicates of 8 × 15 cm plates were prepared from BioID-3×FLAG and IC2-BioID-3×FLAG stable cell lines. Each replicate was lysed in 10 mL 'mild' detergent lysis buffer by mixing gently at 4°C for 20 min. The mixture was centrifuged at 66,000 x g for 30 min in a Type 70 Ti rotor (Beckman Coulter) at 4°C. The lysate was recovered and incubated with 250 µL packed volume anti-FLAG agarose for 16 hr at 4°C with gentle mixing. Beads were collected by centrifugation at 1000 rpm in a microcentrifuge for 2 min and washed four times with 'mild' detergent lysis buffer. Proteins were eluted with 250 µL lysis buffer containing 0.4 mg/mL 3×FLAG peptide at 4°C for 30 min. Eluates were precipitated with TCA as described above. MS/MS analysis was performed as described above.

We used FLAG-immunoprecipitation combined with FPLC to determine the percent incorporation of BioID-tagged subunits into their respective complexes. Cells expressing either IC2-BioID-3×FLAG or p62-BioID-3×FLAG were collected from 8 × 15 cm plates as described above. Cells were lysed in 10 mL GF buffer (50 mM Tris-HCl, pH 8.0; 150 mM NaCl, 10% glycerol, 0.2% Triton X-100, 1 mM DTT, 0.5 mM ATP, 1X protease inhibitor cocktail) and centrifuged as described. The lysate was added to 200 µL packed Anti-FLAG M2 Affinity Gel (Sigma-Aldrich) for 16 hr at 4°C with gentle mixing. After washing in batch twice with 50 mL GF buffer, elutions were performed with 250 µL of lysis buffer with 0.5 mg/mL 3×FLAG peptide at 4°C for 30 min. A Superose 6 Increase 10/300 GL was equilibrated in GF buffer containing 5% glycerol. Molecular weight standards were analyzed first and consisted of a mixture of thyroglobulin (669 kDa), beta-amylase (200 kDa), alcohol dehydrogenase (150 kDa), albumin (66 kDa), and carbonic anhydrase (29 kDa). 200 µL of each BioID fusion protein eluate were then run separately, and 1 mL fractions were collected for each. Selected fractions were mixed with 10 µL packed Anti-FLAG M2 Affinity Gel (Sigma-Aldrich) and incubated with mixing at 4°C for 2 hr. This mixture was then centrifuged briefly, the supernatant was removed, and the resin was boiled in 2X SDS sample buffer. Released proteins were analyzed by Western blotting with anti-FLAG, anti-dynein heavy chain, and anti-p150 dynactin antibodies (see "Western analysis and antibodies, below, for details). Image intensities of bands were quantified using FIJI. Peak anti-FLAG Western band signal intensities of the low molecular weight (= free) and high molecular weight (= incorporated) IC2- and p62-containing species were used to calculate the percent incorporation of IC2 and p62 into the dynein and dynactin complexes, respectively (*Figure 1E and G*). The percent

incorporation was calculated as $\text{intensity}_{\text{incorporated}}/(\text{intensity}_{\text{free}}+ \text{intensity}_{\text{incorporated}})$. To construct the graphs in *Figure 1E* and *Figure 1G* the intensity of each fraction was divided by the sum of the intensities for all fractions and plotted against the elution volume.

## Western analysis and antibodies

Lysates and eluates were run on 4–12% polyacrylamide gels (NuPage, Invitrogen/ Thermo Fisher Scientific) and transferred to PVDF (Immobilon-P, EMD Millipore; Billerica, MA) for 1.5 hr at 300 mA constant current. Blots were blocked for 10 min with TBST +5% dry milk (w/v), and immunoblotted with appropriate antibodies. All antibodies were diluted in TBST +5% milk (w/v). Primary antibodies were incubated overnight at 4°C, while secondary antibodies were incubated for 1 hr at room temperature. Antibodies used were anti-FLAG conjugated HRP (A8592, Sigma-Aldrich, 1:5000 dilution), rabbit anti-dynein heavy chain (R325, Santa Cruz Biotechnology, Dallas, TX, 1:500-1:1000 dilution), mouse anti-p150 dynactin (610474, BD Biosciences, San Jose, CA, 1:500-1:1000 dilution), rabbit anti-BICD2 (ab117818, Abcam; Cambridge, MA, 1:5000 dilution), rabbit anti-FAM160A2 (ab184160, Abcam, 1:1000), rabbit anti-KIF1C (NB100-57510, Novus Biologicals, Littleton, CO, 1:1000), goat anti-rabbit HRP (sc-2030, Santa Cruz Biotechnology, 1:5000 dilution) and goat anti-mouse HRP (sc-2031, Santa Cruz Biotechnology, 1:5000 dilution). Westerns were visualized with Supersignal West Pico or Femto Chemiluminescent reagents (Thermo Fisher Scientific) and a VersaDoc imaging system (Bio-Rad Laboratories; Hercules, CA). Image intensity histograms were adjusted and images were converted to 8-bit with FIJI before being imported into Adobe Illustrator to make figures.

## Secondary screen and hit validation

For the secondary screen described in *Figure 4*, we pooled BioID datasets from the core dynein/ dynactin subunits where at least one known activator (BICD1, BICD2, HOOK1, or HOOK3) was enriched. Proteins were selected for coiled coil analysis if they had a $\text{dNSAF}_{\text{ORF-BioID}}:\text{dNSAF}_{\text{BioID}}$ ratio greater than 3, were present in 3 of 4 replicates, and contained a predicted coiled coil of at least 100 aa. Predicted coiled coil sequences were extracted from UniProt; those from nuclear proteins, dynein/dynactin subunits, and a single protein that was entirely coiled coil (GOLGA4), were discarded. Each coiled coil was then codon optimized for synthesis and expression in mammalian cells, synthesized (IDT), and cloned by isothermal assembly into pcDNA5/FRT/TO as fusions with super folder (sf) GFP (e.g. pcDNA5/FRT/TO-sfGFP-CC$_x$-3×FLAG). A negative control was used consisting of sfGFP-3×FLAG alone, and two known activator coiled coil constructs were used as positive controls (sfGFP-BICD2 [1-422]−3×FLAG, sfGFP-HOOK3 [1-552]−3×FLAG). Transient transfections, 'mild' detergent immunoprecipitations and Western blot analysis were performed as described above.

The length of expression constructs for the positive hits (NINL, daple and girdin), and NIN, a protein closely related to NINL were further optimized (*Figure 4C and D*). Although our initial girdin construct (sfGFP-girdin [1-425]−3×FLAG) immunoprecipitated dynein/dynactin, it was expressed as a truncated protein (open triangle, *Figure 4D*). Informed by a recent study of Hook proteins (both daple and girdin contain a Hook domain) (*Schroeder and Vale, 2016*), we made longer girdin and daple constructs (1–542 and 1–545, respectively). Since girdin was truncated, yet still produced a FLAG positive Western signal, we reasoned that GFP was proteolytically cleaved from the construct in cells. To circumvent this, we moved the sfGFP module to the C-terminus of girdin (girdin [1-542]-sfGFP-3×FLAG). This construct was not proteolyzed, as was the case with a longer version of daple (sfGFP-daple [1-545]−3×FLAG); both constructs immunoprecipitated dynein/dynactin (*Figure 4D*). In our secondary screen NINL (373-702) immunoprecipitated dynein/dynactin (*Figure 4—figure supplement 1A*), while a construct from the closely related protein, NIN (353-580), did not. Based upon a previous report (*Casenghi et al., 2005*), we generated longer versions of NIN (1-693) and NINL (1-702); both immunoprecipitated dynein/dynactin (*Figure 4C*).

sfGFP-tagged HOOK3 constructs were used to map the interaction between HOOK3 and KIF1C (*Figure 6—figure supplement 2*). Full length HOOK3 (1–718) was moved from pcDNA5/FRT/TO-HOOK3-BioID-3×FLAG to pcDNA5/FRT/TO-sfGFP-3×FLAG by PCR and isothermal assembly. pcDNA5/FRT/TO-HOOK3 [1-552]−3×FLAG is described above. pcDNA5/FRT/TO-sfGFP-HOOK3 [553-718] was generated by PCR and ligation using pcDNA5/FRT/TO-sfGFP-HOOK3 [1-718]−3×FLAG as a template.

## Protein purification

Dynein and dynactin were purified from stable HEK293 cell lines expressing IC2-SNAPf-3xFLAG or p62-HALO-3xFLAG, respectively. Cell lines were constructed using the FLP/FRT system (Thermo Fisher Scientific) as outlined above. Between 60–100 80% confluent, 15 cm plates were harvested per purification. Cells were collected by pipetting with ice-cold PBS and centrifuged at 1000 x g for 2 min to pellet. Cells were washed once more with ice-cold PBS. Cell pellets were either snap-frozen in liquid nitrogen in 50 mL conical tubes or immediately lysed for protein purification. To lyse, cell pellets were resuspended in dynein lysis buffer (25 mM HEPES pH 7.4, 50 mM KOAc, 2 mM MgOAc, 1 mM EGTA, 10% glycerol (v/v), and 1 mM DTT) supplemented with 0.2% Triton X-100, 0.5 mM Mg-ATP, and 1X protease inhibitors (cOmplete Protease Inhibitor Cocktail, Roche). To ensure complete lysis, resuspended cells were slowly rotated lengthwise at 4°C for 15 min. The lysate was clarified via centrifugation at 66,000 x g for 30 min in a Type 70 Ti rotor (Beckman Coulter) at 4°C. The clarified supernatant was mixed with 0.75–1 mL of Anti-FLAG M2 Affinity Gel (Sigma-Aldrich) overnight at 4°C. During incubation, the slurry was rotated about its long axis in a full 50 mL Falcon tube. Beads were collected by gravity flow and washed with 50 mL wash buffer (dynein lysis buffer with 0.02% Triton X-100 and 0.5 mM Mg-ATP) supplemented with protease inhibitors (cOmplete Protease Inhibitor Cocktail, Roche). Beads were then washed with 50 mL high salt wash buffer (25 mM HEPES, pH 7.4, 300 mM KOAc, 2 mM MgOAc, 10% glycerol, 1 mM DTT, 0.02% Triton X-100, 0.5 mM Mg-ATP, and 1X protease inhibitor (cOmplete Protease Inhibitor Cocktail, Roche) and then with 100 mL wash buffer.

To label with a fluorophore the beads were resuspended in 1 mL wash buffer and incubated with either 5 µM SNAP-Cell TMR Star (New England BioLabs; Ipswich, MA) (to label IC2) or 5 µM Halo-Atto647N (Promega) (to label p62) for 10 min at room temperature. Unreacted dye was removed from beads with 50–80 mL of wash buffer. Protein complexes were eluted with 0.5–1 mL of elution buffer (wash buffer with 2 mg/mL 3xFLAG peptide). Elution was collected, diluted to 2 mL in Buffer A (50 mM Tris pH 8, 2 mM MgOAc, 1 mM EGTA, and 1 mM DTT) and injected onto a MonoQ 5/50 GL column (GE Healthcare Life Sciences) at 0.5 mL/min. The column was washed with 20 CV of Buffer A at 1 mL/min. To elute, a linear gradient was run over 40 CV into Buffer B (50 mM Tris pH 8, 2 mM MgOAc, 1 mM EGTA, 1 mM DTT, 1 M KOAc). Pure dynein complex elutes from ~60–70% Buffer B, while pure dynactin complex elutes around ~70–80% Buffer B. Peak fractions were pooled and concentrated, Mg-ATP was added to 0.1 mM and glycerol was added to 10%. Samples were then snap frozen in 2 µL aliquots.

Activators and potential activators were cloned into pET-28a vectors with an N-terminal StrepII-sfGFP tag. Mouse BICD2 (mBICD2) (aa 25–400) was a gift from Rick McKenney (University of California, Davis), while NIN (aa 1–693) and NINL (aa 1–702) were sub-cloned from ORFs outlined above. All constructs were transformed into BL21-CodonPlus (DE3)-RIPL cells (Agilent Technologies; Santa Clara, CA). 2 L of cells were grown at 37°C in LB media to a 600 nm optical density of 0.4–0.8 before the temperature was reduced to 18°C and expression was induced with 0.5 mM IPTG. After 16–18 hr, cells were harvested via centrifugation for 6 min at 4°C at 6000 rpm in a JLA 8.1000 fixed angle rotor (Beckman Coulter). Pellets were resuspended in 30–40 mL of dynein lysis buffer with 0.5 mM PefaBloc SC (Sigma-Aldrich) and 1 mg/mL lysozyme and incubated at 4°C for 30 min. Cells were lysed via sonication (Branson Digital Sonifier, Emerson; Saint Louis, MA) and clarified via centrifugation at 66,000 x g for 30 min in a Type 70 Ti rotor (Beckman) at 4°C. Supernatant was loaded onto a 5 mL StrepTrap column (GE Healthcare Life Sciences) and washed with 50–100 mL of lysis buffer. Activators were then eluted with 25–50 mL of elution buffer (dynein lysis buffer with 3 mM d-Desthiobiotin). Finally, all activators were purified via size exclusion chromatography on either a Superdex 200 Increase 10/300 GL or a Superose 6 Increase 10/300 GL column (GE Healthcare Life Sciences) that had been equilibrated with degassed dynein lysis buffer. Peak fractions were collected and used for single molecule motility experiments immediately or snap-frozen in 2–20 µL aliquots. Care was taken not to concentrate the activators as we observed that this led to aggregation and inactivity.

## Single-molecule motility assays and data analysis

Two types of single-molecule motility assays were performed. Owing to the presence of sfGFP on each construct from the secondary screen described above, we were able to use a TIRF-based

motility assay to determine if dynein/ dynactin present in coiled coil immunoprecipitations was activated. Here, immunoprecipitation eluate was imaged in a single-molecule motility assay (see 'Immunoprecipitation' section above for sample preparation details). In this experiment, microtubules were labeled with HiLyte 647 tubulin (Cytoskeleton, Inc.; Denver, CO) for visualization.

In the second type of single-molecule assay, purified ~6 nM dynein (labeled with TMR),~60 nM dynactin (labeled with Atto-647N) and ~24–260 nM bacterially expressed and purified activator (labeled with sfGFP) were mixed together for ten minutes at 4°C before imaging. Immediately before imaging, the dynein/ dynactin/ activator complexes were diluted 1:20-1:80 in imaging buffer (see below for composition). In this experiment, microtubules were labeled with Alexa Fluor 405 tubulin (Thermo Fisher Scientific).

Each type of single-molecule motility assay was performed in flow chambers. Biotinylated and PEGylated coverslips (Microsurfaces; Englewood, NJ) were used to reduce non-specific binding. Microtubules contained ~10% biotin-tubulin for attachment to streptavidin-coated cover slip and ~10% HiLyte 647 tubulin (Cytoskeleton, Inc.) or ~10% Alexa Fluor 405 (Thermo Fisher Scientific) tubulin for visualization. The imaging buffer used consisted of dynein lysis buffer (25 mM HEPES pH 7.4, 50 mM KOAc, 2 mM MgOAc, 1 mM EGTA, 10% glycerol (v/v), and 1 mM DTT) supplemented with 0.75–1 mg/mL casein, 1 mM Mg-ATP, 71.5 mM $\beta$ME (beta-mercaptoethanol) and an oxygen scavenger system (0.4% glucose, 45 µg/ml glucose catalase (Sigma-Aldrich), and 1.15 mg/ml glucose oxidase (Sigma-Aldrich)). Images were recorded every 0.5 s for 10 min. Each individual sample was imaged no longer than 35 min. At least two technical replicates were performed for all single-molecule experiments shown in *Figure 4* and *Figure 5*.

Motility assays were performed with an inverted Nikon (Melville, NY) Ti-E Eclipse microscope equipped with 100 × 1.4 N.A. oil immersion Plano Apo Nikon objective. The xy position of the stage was controlled by ProScan linear motor stage controller (Prior Scientific; Rockland, MA). The microscope was equipped with an MLC400B laser launch (Agilent Technologies) equipped with 405 nm (30 mW), 488 nm (90 mW), 561 nm (90 mW), and 640 nm (170 mW) laser lines. The excitation and emission paths were filtered using appropriate filter cubes (Chroma; Bellows Falls, VT). The emitted signals were detected with an iXon Ultra electron multiplier CCD camera (Andor Technology; United Kingdom). Illumination and image acquisition is controlled by NIS Elements Advanced Research software (Nikon).

The velocity of moving particles was calculated from kymographs generated in ImageJ as described (*Roberts et al., 2014*). For the immunoprecipitation followed by single-molecule experiments, particles moving in the 488 channel (activator) were used for velocity calculations. For the motility experiments with purified components, the 561 channel (dynein) was used for quantification of velocity. Velocities were only calculated from molecules that moved processively for greater than five frames. Non-motile or diffusive events were not considered in velocity calculation.

Processive events were defined as events that move uni-directionally and do not exhibit directional changes greater than 600 nm. Diffusive events were defined as events that exhibit at least one bi-directional movement greater than 600 nm in each direction. Single-molecule movements that change apparent behavior (e.g. shift from non-motile to processive) were counted as multiple events.

## Acknowledgements

We thank Chris Patil, Agnieszka Kendrick, John Salogiannis, and Andres Leschziner for critical comments on the manuscript, Kyle Roux for sharing the G2 BioID plasmid ahead of publication, and Wade Harper and members of the Harper lab for scientific advice. MPW, LAF and SES are supported by the Stowers Institute for Medical Research. SRP is supported by the NIH (R01GM121772 and R01GM107214) and is a Howard Hughes Medical Institute-Simons Faculty Scholar. MED is a Jane Coffin Childs Fellow and ZMH is supported by a NSF graduate fellowship.

## Additional information

### Competing interests

SLR-P: Reviewing editor, *eLife*. The other authors declare that no competing interests exist.

## Funding

| Funder | Grant reference number | Author |
|---|---|---|
| National Institutes of Health | R01GM121772 | William B Redwine<br>Phuoc Tien Tran<br>Samara L Reck-Peterson |
| Howard Hughes Medical Institute | Faculty Scholar | Morgan E DeSantis<br>Samara L Reck-Peterson |
| Simons Foundation | Faculty Scholar | Morgan E DeSantis<br>Samara L Reck-Peterson |
| National Institutes of Health | R01GM107214 | Ian Hollyer<br>Samara L Reck-Peterson |
| Jane Coffin Childs Memorial Fund for Medical Research | | Morgan E DeSantis |
| National Science Foundation | | Zaw Min Htet |
| Stowers Institute for Medical Research | | Selene K Swanson<br>Laurence Florens<br>Michael P Washburn |

The funders had no role in study design, data collection and interpretation, or the decision to submit the work for publication.

## Author contributions

WBR, Conceptualization, Data curation, Formal analysis, Supervision, Validation, Investigation, Visualization, Methodology, Writing—original draft, Writing—review and editing; MED, Conceptualization, Data curation, Formal analysis, Supervision, Funding acquisition, Validation, Investigation, Visualization, Methodology, Writing—original draft, Writing—review and editing; IH, Formal analysis, Validation, Methodology; ZMH, Formal analysis, Validation, Investigation, Visualization, Methodology; PTT, Investigation, Methodology; SKS, Resources, Data curation, Software, Formal analysis, Methodology; LF, Resources, Data curation, Software, Formal analysis, Supervision, Project administration; MPW, Supervision, Funding acquisition, Project administration; SLR-P, Conceptualization, Resources, Supervision, Funding acquisition, Visualization, Writing—original draft, Project administration, Writing—review and editing

## Author ORCIDs

William B Redwine, http://orcid.org/0000-0001-6342-7253
Morgan E DeSantis, http://orcid.org/0000-0003-4761-3691
Michael P Washburn, http://orcid.org/0000-0001-7568-2585
Samara L Reck-Peterson, http://orcid.org/0000-0002-1553-465X

## Additional files

### Supplementary files

• Supplementary file 1. Master file of mass spectrometry data related to *Figures 1*, *2*, *3* and *6*. This excel file contains all of the mass spectrometry data referenced in the manuscript. The first blue tab corresponds to the Flag immunoprecipitation experiment in *Figure 1B*. The second blue tab contains the dynein core subunits detected in the Flag immunoprecipitation experiment, corresponding to *Figure 1C*. The purple tab contains all mass spectrometry data related to *Figure 2*. The pink tabs contain all of the dynein/dynactin interactome mass spectrometry data. This data was used to generate *Figure 3C*. The green tabs contain all of the activator interactome mass spectrometry data. This data was used to generate *Figure 6* and *Figure 6—figure supplement 1*. 'NT' and 'CT' indicate that the BioID tag was on the N-terminus or C-terminus of the full-length protein, respectively.

• Supplementary file 2. Mass spectrometry data related to *Figure 2*. This excel file contains the mass spectrometry data that was used to generate *Figure 2C* (second tab) and 2D (first tab).

• Supplementary file 3. Mass spectrometry data related to *Figure 3C* and *Figure 3—figure supplement 1*. The blue tabs contain the BioID interactome data for IC1, IC2, LIC1, LIC2, TcTex, RB, and p62. Only the data for dynein and dynactin subunits and known (BICD2, HOOK1 and HOOK3) and suspected (BICD1) activators are shown. The blue tab titled 'mapping color code' lists the dynein and dynactin subunits enriched in the BioID experiments and graphically displayed in *Figure 3C*. Shading indicates enrichment value: light gray (2–3 fold) or dark gray ($\geq$3 fold), p<0.05 (Student's two-tailed t-test). The entire datasets can be found in *Supplementary file 1* (pink tabs in *Supplementary file 1*). The pink tabs in this excel file contain all of the significant hits from each BioID tagged dynein and dynactin subunit. Significance was defined as >3 fold enrichment, p-values<0.05 (Students two-tailed t-test), and average spectral counts > 2. This data was used to generate the network shown in *Figure 3—figure supplement 1*. The pink tab titled 'core hits' lists the gene names for all hits, specific hits (unique to each tagged subunit), and hits shared by at least two datasets, for the dynein and dynactin BioID tagged subunits. The pink tab titled 'core Venn' contains the output from the Venn analysis (http://bioinformatics.psb.ugent.be/webtools/Venn/) of the dynein and dynactin core subunit interactomes used to generate the network shown in *Figure 3—figure supplement 1*. Proteins found in only one dataset are listed in the excel file, but not shown in the network.

• Supplementary file 4. Mass spectrometry data related to *Figure 6* and *Figure 6—figure supplement 1*. The green tabs contain all significant hits from the NT and CT BioID tagged activator datasets. Significance was defined as >3 fold enrichment, p-values<0.05 (Students two-tailed t-test), and average spectral counts > 2. The blue tab titled 'NT hits' lists the gene names for all hits, specific hits (unique to each tagged activator), and hits shared by at least two datasets, for the NT-activator BioID tagged subunits. The blue tab titled 'NT Venn' contains the output from the Venn analysis (http://bioinformatics.psb.ugent.be/webtools/Venn/) of the NT activator interactomes used to generate the network shown in *Figure 6B*. Proteins found in only one dataset are listed in the excel file, but not shown in the network. The blue tab titled 'white spheres NT' highlights (in grey) the four hits that were shared by an activator from each activator family (i.e. BICD, HOOK, and NIN). The orange tab titled 'CT hits' lists the gene names for all hits, specific hits (unique to each tagged activator), and hits shared by at least two datasets, for the CT-activator BioID tagged subunits. The orange tab titled 'CT Venn' contains the output from the Venn analysis (http://bioinformatics.psb.ugent.be/webtools/Venn/) of the CT activator interactomes used to generate the network shown in *Figure 6B*. Proteins found in only one dataset are listed in the excel file, but not shown in the network. The orange tab titled 'white spheres CT' highlights (in grey) the 21 hits that were shared by an activator from each activator family (i.e. BICD, HOOK, and NIN). The pink tabs contain data that contrast all of the activator hits, combining the NT and CT datasets. The pink tab titled 'NT-CT combined hits' contains the gene names that are specific for each termini of each activator or shared between any dataset. The pink tab titled 'NT-CT combined Venn' contains the output from the Venn analysis (http://bioinformatics.psb.ugent.be/webtools/Venn/) of all activator interactomes used to generate the network shown in *Figure 6—figure supplement 1*.

• Supplementary file 5. GO analysis of dynein activator C-terminal BioID datasets. This excel file contains gene ontology analyses using the 'cellular component' option (GO, geneontology.org). The GO terms with p-values<0.05 are shown for hits that were shared in at least three C-terminal BioID datasets; C-terminal hits that were specific to BICD2, HOOK1, HOOK3, NIN, and NINL; and C-terminal hits that were shared by activator family members BICD1 and BICD2, HOOK1 and HOOK3, and NIN and NINL.

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
