## [Decision Letter]

Thank you for submitting your article "The human cytoplasmic dynein interactome reveals novel activators of motility" for consideration by *eLife*. Your article has been favorably evaluated by Vivek Malhotra (Senior Editor) and three reviewers, one of whom is a member of our Board of Reviewing Editors. The reviewers have opted to remain anonymous.

The reviewers have discussed the reviews with one another and the Reviewing Editor has drafted this decision to help you prepare a revised submission.

The referees enthusiastically support the new approach of BioID you are taking to uncover the Dynein interactome as well as to identify potential dynein interactors.

The referees believe that the paper would be more appropriate for the "tools and resources" section since the biological mechanistic insight from the study is somewhat limited. I hope you can recast your story along these lines.

The referees have raised a number of points that I would like you to consider, but perhaps the most important of them is the point 2 of referee 1 and point 1 of referee 2, who would like to see perhaps another line of evidence that the interactions are physiological. Perhaps co-IPs and / orcofractionation could be an approach you can explore. Such a validated database would provide a solid platform for the field to move forward in this exciting area.

Most other points raised are rather minor in nature and I would like you to consider them and respond with appropriate changes to the text or experiments (if already available).

The referees’ comments are provided verbatim for your information.

*Reviewer #1:*

This manuscript by Reck-Peterson and colleagues used the BioID strategy to identify proteins that transiently interact and are / or located within ~ 10 angstroms of dynein or its associated proteins. The work produces a very impressive data set of protein interactions. They use the presence of coiled-coil domains and ability to coimmunoprecipitate as criteria to focus on a few proteins as potential activators. They show that ninein and ninein-like are strong candidates to act as activators of dynein-based motility. In general the work is very well done and I have three major points that need to be considered before publication.

1) I feel this paper fits a "resource and tools" article more than a research paper,since despite the large and extremely useful datasets arising from the work, any actual mechanistic insight is limited. I don't expect the authors to investigate mechanism of dynein activation by ninein or ninein-like.

2) For this to be useful as a dataset, is there a way the authors can test how many of these interactions happen under physiological conditions using co-IPs or even better using short-range cross-linkers.

3) Figure 4 motility assays seem to miss some control. The experiments have a positive control of known activators, but no negative control. I am not sure what would constitute a negative control of dynein without activators. There is a kind of control incorporated in Figure 5, which may be more appropriate in Figure 4, or a better control is needed in Figure 4.

*Reviewer #2:*

This manuscript describes efforts to identify interactors of the dynein complex using the BioID method. The authors produce a very large dataset as a result of BioID G2-tagging of several dynein subunits and p62 of dynactin, as well as known coiled-coil activators of dynein-dynactin motility. The dataset will be of great interest to the dynein field, as previous proteomic studies have used methods that are likely to miss transient protein-protein interactions. In addition to other dynein-dynactin subunits, a small number of proteins are identified in proximity to multiple dynein subunits, which will undoubtedly seed future mechanistic studies. In this study, the authors follow up NIN and NINL, which were identified in the IC1 and IC2 BioID experiments, and were previously shown by others to associate with dynein and dynactin in cells. It is convincingly shown that NIN and NINL are sufficient to activate dynein-dynactin motility in vitro, thus increasing the number of known activators of dynein. The authors also identify a large number of proteins in proximity to the C-terminal regions of previously known coiled-coil activators of dynein, as well as NIN and NINL. The long list contains some known cargo adaptors that were shown to associate with the activators by more stringent methods, suggesting that it also contains novel proteins of this class. The authors use this information to hypothesize that activators/adaptors have many interaction partners, constituting a tiered mechanism for linking specific cargoes to dynein.

Overall the experiments are well controlled and the advances are substantive. However, there are some essential points that need to be addressed.

1) For a publication in a journal of this standing I would expect to see evidence that the list of proteins identified in proximity to the C-terminal regions of activators contains novel, bona fide interactors. An alternative possibility is that these proteins do not interact directly with the activators but are present at high concentrations in the regions of the cells where these fusion proteins accumulate. Without some validation readers will be unsure how much confidence they can have in this part of the dataset. This issue could, for example, be addressed by performing immunoprecipitations from tissue culture cells for at least one of the C-terminal constructs, followed by western blotting with available antibodies to some of the proteins on the list. Whereas some bona fide interactions may be too transient to detect by this method, others may not be. Elucidating which biological processes depend on validated interactions is not, however, necessary.

2) The identification of NIN and NINL as activators of dynein is a key part of the study and the authors should state explicitly in the Results section of the text in which BioID experiments they were (and were not) identified.

3) The legends for the motility assays should describe if the number of complexes analyzed come from a single technical replicate or a combination of multiple technical replicates. The Methods should give more information on what constitutes a technical replicate in this study.

*Reviewer #3:*

Cytoplasmic dynein is a microtubule motor that transports many different cargoes. While recent work has revealed how dynein is activated by binding an adaptor protein and dynactin to form a tripartite complex, what is not clear is how many adaptors there are, and whether there is one per cargo, and so forth. This work provides significant new insight into this question, by using the proximity ligation BioID approach combined with mass spectrometry. Two new adaptors, NIN and NINL, are identified as interacting with dynein and dynactin. Importantly, the authors also clearly demonstrate that this interaction leads to activation of dynein's motor activity, using elegant in vitro motility approaches. The authors go on to perform BioID on the new and previously identified adaptors themselves, which reveals a range of shared and distinct interacting proteins. This leads to the conclusion that there is a small number of adaptors, each of which can interact with multiple cargo components. One surprise is that several known membrane-associated adaptors were not detected (RILP, Rab11FIP3), but this is explained by their low expression in HEK cells. A complication of the approach used is that while ~50% of the BirA-tagged IC2 or p62 subunits are in the dynein or dynactin complexes, respectively, that does mean that 50% are not, and could be contributing non-specific interactions. Even with this caveat, this is an important study.

Overall, the work is thorough, well presented and explained, and the conclusions are sound. The methods are described in detail, which is excellent to see. The mass spectrometry data will be a valuable resource for the motor community. I have no substantive concerns.

---

## [Author Response]

*[…] Reviewer #1:*

*This manuscript by Reck-Peterson and colleagues used the BioID strategy to identify proteins that transiently interact and are / or located within ~ 10 angstroms of dynein or its associated proteins. The work produces a very impressive data set of protein interactions. They use the presence of coiled-coil domains and ability to coimmunoprecipitate as criteria to focus on a few proteins as potential activators. They show that ninein and ninein-like are strong candidates to act as activators of dynein-based motility. In general the work is very well done and I have three major points that need to be considered before publication.*

*1) I feel this paper fits a "resource and tools" article more than a research paper,since despite the large and extremely useful datasets arising from the work, any actual mechanistic insight is limited. I don't expect the authors to investigate mechanism of dynein activation by ninein or ninein-like.*

Thank you for the suggestion. We will resubmit this paper as a “resource and tools” article.

*2) For this to be useful as a dataset, is there a way the authors can test how many of these interactions happen under physiological conditions using co-IPs or even better using short-range cross-linkers.*

To include new validation data for our activator datasets we have now added a new figure (Figure 6—figure supplement 2) investigating a unique interaction found in the HOOK1 and HOOK3 BioID datasets. We chose to focus on these datasets because we were intrigued by the presence of the kinesin, KIF1C, in both the HOOK1 and HOOK3 datasets, but not in the dataset of any other activator. KIF1C is a kinesin-3 and its interaction with a dynein activator suggests that HOOK proteins could serve as scaffolds for plus- and minus-end-directed motors. Our results are summarized in the following paragraph.

We used co-immunoprecipitation experiments to verify the interaction between KIF1C and HOOK3 (new Figure 6—figure supplement 2). In addition, we showed that this interaction requires the C-terminus of HOOK3 (new Figure 6—figure supplement 2). Highlighting the power of BioID to detect proximal interactions, we only identified KIF1C in the HOOK1 and HOOK3 C-terminal BioID datasets([Supplementary-material SD4-data]). Conversely, dynein and dynactin were enriched only in the N-terminal BioID datasets and dynein was immunoprecipitated only by full length and N-terminal portions of HOOK3 (new Figure 6—figure supplement 2 and [Supplementary-material SD4-data]). Together, these results further validate our approach and provide additional support for the spatially restricted nature of BioID. They also provide an exciting new biological finding that HOOK proteins may function as scaffolds for both plus- and minus-end-directed motors. We have added text in both the Results and Discussion to highlight this important finding.

We would also like to emphasize that BioID involves the labeling of proteins within cells, in their physiological environments – the cytoplasm for constructs used in our study. Any experiment performed post-lysis can only be an approximation of the conditions that exist in a living cell. As mentioned by reviewer #2, BioID also has the potential to identify interactions that are transient or of low affinity; thus, BioID can identify interactions that do not survive the dilution or the timescale of an immunoprecipitation experiment. In addition to the new experiment in Figure 6—figure supplement 2 wealth of data in our original submission validated the power of our approach (see points 1-4 below). We feel that further validation would best be done using localization and depletion studies in living cells, but think that this is beyond the scope of the current work.

1) In the dynein and dynactin interactomes we identified known subunits of the dynein and dynactin complexes that are known to be in close proximity to the labeled subunit based on cryo-EM structures (Figure 2, Figure 3, Figure 3—figure supplement 1, and [Supplementary-material SD1-data]) (Chowdhury et al., 2015; Urnavicius et al., 2015).

2) We identified known (BICD2, HOOK1 and HOOK3) or suspected (BICD1) activators in the datasets of dynein or dynactin subunits that are located in close proximity to activators based on cryo-EM structures (Figure 4 and Figure 4—figure supplement 1 and [Supplementary-material SD1-data]) (Chowdhury et al., 2015; Urnavicius et al., 2015).

3) We performed immunoprecipitation experiments using 24 coiled coil-containing proteins that were enriched in our dynein/dynactin interactome. Four of these proteins co-immunoprecipitated with dynein and dynactin (NIN, NINL, CCDC88A, and CCDC88C). While we did not detect a dynein/dynactin interaction with the other 20 proteins by co-immunoprecipitation, as described above, this could be because the interactions are not stable enough to be detected under the conditions we use for an immunoprecipitation experiment. It is also possible that some of these “hits” do not represent bona fide interactions with dynein/dynactin. Localization experiments in cells, which we think are beyond the scope of this work, will be the best next step to verify these interactions.

4) We identified a number of known interactions in our activator datasets, further validating our approach (for more details see Discussion).

a) AKTIP/FTSis a known HOOK1 and HOOK3 interacting protein present in our HOOK datasets.

b) FAM160A2/FHIP is a known HOOK1 and HOOK3 interacting protein in our HOOK datasets. We also identified two homologues of FHIP in our datasets: FAM160A1 and FAM160A2.

c) RIMBP3 is a known HOOK1-interacting protein that we identified in our dataset.

d) RANBP2 is a well-characterized BICD2 interacting protein that we identified in our BICD2 dataset.

e) MICAL3 is a known interaction partner of NINL, which we identified in our NINL dataset.

*3) Figure 4 motility assays seem to miss some control. The experiments have a positive control of known activators, but no negative control. I am not sure what would constitute a negative control of dynein without activators. There is a kind of control incorporated in Figure 5, which may be more appropriate in Figure 4, or a better control is needed in Figure 4.*

We have included a kymograph of the control immunoprecipitation of soluble GFP, which shows no motile events (new Figure 4—figure supplement 2). Another informative control/comparison is our daple and girdin data, which show that although daple and girdin can bind dynein/dynactin, motile events are extremely rare (new Figure 4—figure supplement 2). Motile events are also extremely rare with fully reconstituted dynein/dynactin (Figure 5), as has been reported previously for dynein/dynactin in the absence of an activator (McKenney et al., 2014; Schlager et al., 2014; Trokter et al., 2012).

*Reviewer #2:*

*[…] Overall the experiments are well controlled and the advances are substantive. However, there are some essential points that need to be addressed.*

*1) For a publication in a journal of this standing I would expect to see evidence that the list of proteins identified in proximity to the C-terminal regions of activators contains novel, bona fide interactors. An alternative possibility is that these proteins do not interact directly with the activators but are present at high concentrations in the regions of the cells where these fusion proteins accumulate. Without some validation readers will be unsure how much confidence they can have in this part of the dataset. This issue could, for example, be addressed by performing immunoprecipitations from tissue culture cells for at least one of the C-terminal constructs, followed by western blotting with available antibodies to some of the proteins on the list. Whereas some bona fide interactions may be too transient to detect by this method, others may not be. Elucidating which biological processes depend on validated interactions is not, however, necessary.*

Please see response #2 to reviewer #1.

*2) The identification of NIN and NINL as activators of dynein is a key part of the study and the authors should state explicitly in the Results section of the text in which BioID experiments they were (and were not) identified.*

Thank you for this suggestion. We have amended the text to include specific reference to the core dynein and dynactin subunit datasets enriched for NIN and NINL.

*3) The legends for the motility assays should describe if the number of complexes analyzed come from a single technical replicate or a combination of multiple technical replicates. The Methods should give more information on what constitutes a technical replicate in this study.*

We have added this information.

*Reviewer #3:*

*[…] Overall, the work is thorough, well presented and explained, and the conclusions are sound. The methods are described in detail, which is excellent to see. The mass spectrometry data will be a valuable resource for the motor community. I have no substantive concerns.*

We thank the reviewer for these comments.